# Path-dependent Discrete Amortized Inference

**Tiago da Silva** [1]  **Esmeralda S. Whitammer** [2 3]  **Salem Lahlou** [1]

## Abstract

We consider the problem of sampling compositional and discrete objects from a given unnormalized posterior distribution. Notably, recent studies have shown that this problem can be efficiently solved by learning a deterministic Markov Decision Process (MDP) that progressively builds each object in proportion to the posterior. In this work, however, we demonstrate that the Markovian assumption can both hamper signal propagation during training and catastrophically reduce the learned sampler's expressivity due to state aliasing. To address these issues, we propose lifting the MDP with a learnable latent dynamical system that allows the underlying policy to depend on the entire past trajectory—and not only on the current state. In view of this, we refer to the resulting method as *path-dependent discrete amortized inference*. Importantly, we provably extend existing learning algorithms for discrete amortized samplers to our setting. In experiments on standard benchmark problems, we also show that our approach often leads to faster learning convergence and improved state space exploration relatively to prior techniques.

## 1. Introduction

When dealing with complex stochastic models, there is only so much one can achieve analytically. Instead, practitioners frequently rely on approximate algorithms for drawing samples from a target distribution, which are used to produce Monte Carlo approximations to a desired functional (Kirkpatrick et al., 1983; Geyer, 1991; Robert et al., 1999; Blei & et al., 2017). For continuous state spaces, this *sampling problem* can often be efficiently addressed by the celebrated Hamiltonian Monte Carlo algorithm (HMC; Neal et al., 2011; Carpenter et al., 2017). Similarly to other Markov

chain Monte Carlo (MCMC) techniques, HMC operates by constructing a Markov chain for which the target distribution is stationary. Nonetheless, its central idea is to augment the state space with a *momentum variable* that evolves through a partly deterministic process dictated by the Hamiltonian dynamics (Hamilton, 1833), enhancing exploration and reducing convergence (mixing) time (Betancourt, 2017).

Apart from continuously relaxing the state space (Maddison et al., 2017; Jang et al., 2017; Han et al., 2020), which can only be done in low-dimensional settings, there has been no corresponding one-size-fits-all solution for the sampling problem for discrete distributions. This issue has been particularly challenging in the presence of compositional objects (e.g., graphs and sequences), as the combinatorial explosion of the induced space often makes generic samplers impractical and forces the reliance on ad hoc techniques (Dinh et al., 2017). To fill this gap, Bengio et al. (2021; 2023) recently introduced a novel class of models—termed Generative Flow Networks (GFlowNets)—that cast the sampling problem as learning a Markov decision process (MDP) that constructs each compositional object according to a given unnormalized distribution. Since then, GFlowNets have been thoroughly studied (Lahlou et al., 2023; Deleu & Bengio, 2023; Shen et al., 2023a; Yu, 2025), with successful applications on phylogenetic inference (Zhou et al., 2024) and causal discovery (Deleu et al., 2022; 2023), among others, that repeatedly demonstrated their competitiveness against ad hoc MCMC-based alternatives. On top of that, their relationship to hierarchical variational inference (Malkin et al., 2023) and reinforcement learning (RL; Tiapkin et al., 2024; Deleu et al., 2024) methods has also been formally established.

From a reinforcement learning perspective, in particular, a GFlowNet can be viewed as solving an MDP problem on a finite and completely observable environment (Sutton et al., 1998; Mohri et al., 2018) defined by the iterative process of constructing the compositional objects (e.g., adding ancestral nodes to a phylogenetic tree). Importantly, as we cannot directly assign unnormalized probabilities—i.e., rewards—to incomplete states (Pan et al., 2023b;a), GFlowNets are often impaired by reward sparsity and a consequent inefficient credit assignment during training (Jang et al., 2024). In addition, although the seminal work of Bengio et al. (2023) demonstrated that there always exists

---

[*]Equal contribution  [1]MBZUAI  [2]University of Edinburgh  [3]CIFAR Fellow. Correspondence to: Tiago da Silva <tiago.dasilva@mbzuai.ac.ae>.

*Proceedings of the 43rd International Conference on Machine Learning*, Seoul, South Korea. PMLR 306, 2026. Copyright 2026 by the author(s).

a deterministic MDP that generates each complete state in proportion to a given target distribution, recent theoretical results indicate that the expressive power of commonly used deep neural networks may be insufficient for representing this optimal policy (Silva et al., 2025a; Kim et al., 2025a).

To mitigate these issues, we propose expanding the given MDP with a history-dependent auxiliary variable. The graphical models for the resulting generative process alongside that of the original GFlowNet are illustrated in Figure 1b. As with HMC, this variable follows a deterministic and state-conditional dynamics defined by a discretized differential equation. (Although it can also be made stochastic; see Section D). Remarkably, as we explain in Section 4, both the vector field underlying the dynamics and the policy function can be jointly learned by stochastic gradient descent on a single loss function. Indeed, from an algorithmic learning standpoint, the introduced latent variable might also be thought of as the memory component of a recurrent neural network (Schmidhuber, 1992; Hochreiter & Schmidhuber, 1997; Irie & Gershman, 2025).

In this context, we refer to our method as *path-dependent discrete amortized inference*. As in hidden Markov models (Baum & Petrie, 1966), by lifting the original state space with a dynamic latent variable, we can no longer ensure that the state-only stochastic process ($\{s_t\}_{t \geq 1}$; blue in Figure 1b) is Markovian. As we argue in Section 3, however, the introduction of path-dependent state-transitions can strictly improve the expressivity of the hypothesis class defined by the space of parametric policies. To show this, we present illustrative environments and distributions that can be solved through path-dependent parameterizations, but not through their Markovian counterparts.

Additionally, traditional learning objectives for GFlowNets (Malkin et al., 2022; Madan et al., 2022; Zhang et al., 2023b; Tiapkin et al., 2024) have to be adjusted to preserve their validity within our lifted space. We carry this out in Section 4 for the trajectory balance (TB), subtrajectory balance (SubTB), and contrastive balance (CB) loss functions. Complementarily, our empirical analysis in Section 5 suggests that the proposed path-dependent algorithm frequently achieves faster learning convergence and a better fit to the target than a similarly parameterized Markovian model. Besides, we also demonstrate in Section E that task-specific techniques for distributed and streaming inference can be seamlessly adapted to the non-Markovian setting (da Silva et al., 2024b). In summary, our contributions are as follows.

1. We introduce a path-dependent algorithm for amortized inference in discrete and compositional spaces, outlining the limitations of the conventional Markovian methods.

2. We provably extend existing learning algorithms for discrete amortized samplers to this new setting, demonstrating their effectiveness on standard benchmark problems.

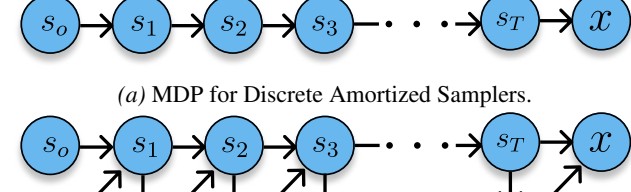

*(a)* MDP for Discrete Amortized Samplers.

*(b)* Lifted MDP.

*Figure 1.* **Graphical model for our algorithm.** We propose attaching a state-conditional latent dynamical system $\{W_t\}_{t=1}^T$ (b) to the GFlowNet's traditional state-only Markov chain $\{s_t\}_{t=1}^T$ (a). The resulting stochastic process, $\{(s_t, W_t)\}_{t=1}^T$, can be described by $s_t | s_{t-1}, W_{t-1} \sim p(\cdot | s_{t-1}, W_{t-1})$ and $W_t = W_{t-1} + \phi(s_t, W_{t-1})$ for $1 \leq t \leq T$ and learnable $p, \phi$ (see Section 4).

## 2. Preliminaries

**Notations.** Let $\mathcal{X}$ be a finite set, and $R \colon \mathcal{X} \to \mathbb{R}_+$ be a positive function defined on $\mathcal{X}$. Our objective is to generate samples from the distribution $\pi(x) \propto R(x)$ over $\mathcal{X}$ induced by $R$. For this, we assume that $\mathcal{X}$ has a *compositional structure*, that is, each $x \in \mathcal{X}$ can be obtained through the addition of simple components to an *initial state* $s_o$. This iterative process characterizes a *pointed DAG*, or a *state graph*, which is central to our work (Bengio et al., 2023).

**Definition 2.1** (Pointed DAG). A *pointed DAG* is a tuple $G = (\mathcal{S}, \mathcal{X}, E)$ composed of *terminal* (or *complete*) states $\mathcal{X}$, (incomplete) *states* $\mathcal{S}$, and directed edges $E \subseteq \mathcal{S} \times (\mathcal{S} \cup \mathcal{X})$. The elements of $\mathcal{X}$ are the only nodes of $G$ without outgoing edges. Also, there is a unique $s_o \in \mathcal{S}$, called *initial state*, without incoming edges and connected to every $x \in \mathcal{X}$ through a directed path (or *trajectory*). See Figure 2.

We say that a trajectory $\tau = (s_t)_{t=0}^T \in \mathcal{S}^{T-1} \times \mathcal{X}$ is *complete* if it starts at the initial state $s_o$ and finishes at $s_T \in \mathcal{X}$.

**GFlowNets.** A GFlowNet (Bengio et al., 2021; Lahlou at al., 2023) learns a Markovian *forward* policy $p_F \colon \mathcal{S} \times (\mathcal{S} \cup \mathcal{X}) \to [0, 1]$ over a state graph such that the marginal of $p_F(s_o, \cdot)$ over the terminal states $\mathcal{X}$,

$$p_\mathsf{T}(x) \coloneqq \sum_{\tau = (s_t)_{t=0}^T \rightsquigarrow x} \prod_{1 \leq t \leq T} p_F(s_t | s_{t-1}), \qquad (1)$$

matches $R(x)$ up to a normalizing constant; $\tau \rightsquigarrow x$ means that $\tau$ is complete and finishes at $x$. In this case, $p_F(\cdot | s) \coloneqq p_F(s, \cdot)$ is supported on the states $s'$ for which $(s, s') \in E$ in the pointed DAG. A GFlowNet also introduces a *backward* policy $p_B$, which is a forward policy on the transposed state graph, that is used for tractable learning of $p_F$. Readers may consult Section C and (Bengio et al., 2023; Malkin et al., 2022) for a comprehensive literature review on GFlowNets.

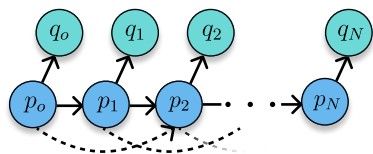

*Figure 2.* Illustration of the LINES environment with length $N$ and step size $M = 2$. Starting at $p_o$, an agent can either move up to $M$ steps forward or transition to a terminal state ($q$) and stop.

## 3. On the limitations of Markovian samplers

This section lays out the empirical and provable limitations of Markovian samplers. In Sections 3.1 and 3.2, we demonstrate that the introduction of path-dependence strictly improves the model's expressivity under certain conditions. In Section 3.3, we experimentally show that the phenomenon of *state aliasing* (Whitehead & Ballard, 1991; Pardo et al., 2022), in which distinct states become representationally indistinguishable from the agent's perspective, is dramatically mitigated by a path-dependent parameterization.

**A running example.** Throughout this section, we consider the simple state graph depicted in Figure 2, which we refer to as the $N$-length LINES environment. There, an agent starts at $p_o$ and, from any state $p_k$, can choose between two actions: either progressing up to $M$ states forward along the main path (i.e., to $p_{k+j}$ for $1 \leq j \leq \min\{M, N-k\}$) or transitioning directly to the terminal state $q_k$ and stopping. The only information available at each $p_k$ and $q_k$ is the state's position $k$. For convenience, we let $\mathcal{P} = \{p_o, p_1, \ldots, p_N\}$ and $\mathcal{Q} = \{q_o, q_1, \ldots, q_N\}$. Under the formalism of Definition 2.1, $\mathcal{S} = \mathcal{P}$ and $\mathcal{X} = \mathcal{Q}$ is the set of terminal states. Clearly, when $M = 1$, the LINES environment corresponds to a one-dimensional version of the standard hypergrid environment (Bengio et al., 2021; Madan et al., 2025). Our objective is to sample each $q_k$ proportionally to a given positive function $R \colon \mathcal{Q} \to \mathbb{R}_+$.

### 3.1. Linear policies

This problem can be addressed by learning a forward policy $p_F \colon \mathcal{P} \times (\mathcal{P} \cup \mathcal{Q}) \to [0, 1]$ satisfying, for a backward policy $p_B$, the trajectory balance condition (Malkin et al., 2022):

$$\frac{p_F(q_{i_k}|p_{i_k:i_o}) \prod_{1 \leq j \leq k} p_F(p_{i_j}|p_{i_{j-1}:i_o})}{p_B(p_{i_k}|q_{i_k}) \prod_{1 \leq j \leq k} p_B(p_{i_{j-1}}|p_{i_j})} = \frac{R(q_{i_k})}{Z} \quad (2)$$

for each increasing sequence $(i_o, \ldots, i_k)$ such that $|i_j - i_{j-1}| \leq M$ and $0 \leq i_j \leq N$ and $0 \leq k \leq N$ and $i_o = 0$, in which $i_j : i_o = (i_o, i_1, \ldots, i_j)$ for $0 \leq j \leq k$. Although a sufficiently wide (Cybenko, 1989; Hornik et al., 1989) or deep (Lu et al., 2017; Hanin & Sellke, 2018) neural network can approximate any smooth function, including a policy abiding by Equation (2), such a model is not mathematically tractable (Laurent & Brecht, 2018).

Instead, motivated by the literature on deep linear networks (Baldi & Hornik, 1989; Baldi & Lu, 2012; Saxe et al., 2014; Laurent & Brecht, 2018) and inverse reinforcement learning (Arora & Doshi, 2020; Metelli et al., 2023), we consider linearly parameterized policies in order to understand the differences between the Markovian and path-dependent approaches for discrete amortized inference. That is, we fix a *state-embedding* function $\psi \colon \mathcal{P} \to \mathbb{R}^d$ for intermediate states and examine policy functions with the form

$$p_F(p_{i+k}|p_i, \mathbf{W}_i) = \text{Softmax}((\mathbf{W}_i \psi(p_i)) \odot \mathbf{m}_i)_k \quad (3)$$

for $1 \leq k \leq M$, $\mathbf{W}_i \in \mathbb{R}^{d \times (M+1)}$ as a learnable parameter and $\mathbf{m}_i \in \{1, -\infty\}^{M+1}$ as a mask that inhibits actions leading to positions with index larger than $N$. Under a conventional Markovian linear model, $\mathbf{W}_i = \mathbf{W}$ is fixed. This represents a standard GFlowNet implementation (Viviano et al., 2025). In this scenario, we say that an unnormalized distribution $R \colon \mathcal{Q} \to \mathbb{R}_+$ is *learnable* by our sampler if there exists a $\mathbf{W}$ for which the marginal distribution of $p_F(\cdot|s_o)$ on $\mathcal{Q}$ matches $R$ up to a normalizing constant (see (1)). Notably, when $\{\mathbf{W}_t\}_{t \geq 1}$ follows a linear dynamics,

$$\begin{aligned} \mathbf{W}_t &= \mathbf{b} \mathbf{a}_t^T, \\ \mathbf{a}_t &= \mathbf{a}_{t-1} + (\mathbf{w}_a^T \psi(s_t)) \cdot \mathbf{v}_a, \end{aligned} \quad (4)$$

with $\mathbf{a}_o \in \mathbb{R}^d$, $\mathbf{w}_a \in \mathbb{R}^d$, and $\mathbf{v}_a \in \mathbb{R}^d$ as learnable parameters and $\mathbf{b} \in \mathbb{R}^{M+1}$ fixed normally at random, then the space of learnable distributions under Equation (4) is strictly larger than that of a corresponding Markovian model; this is the content of the next proposition.

**Proposition 3.1.** *Let $M = 1$ and $\mathbf{r} = (R(q_i))_{i=0}^N$ be the vector of state-wise probabilities. Clearly, both $\lambda \mathbf{r}$ and $\mathbf{r}$ induce the same distribution over $\mathcal{Q}$ for $\lambda > 0$; let $[\mathbf{r}]$ be the corresponding equivalence class. Then, define*

$$\mathcal{W}_M = \{[\mathbf{r}] \colon \exists \mathbf{W} \text{ s.t. } p_{\mathsf{T}}(q_i) \propto [\mathbf{r}]_i\}$$
$$\text{and } \mathcal{W}_P = \{[\mathbf{r}] \colon \exists \mathbf{a}_o, \mathbf{w}_a, \mathbf{v}_a \text{ s.t. } p_{\mathsf{T}}(q_i) \propto [\mathbf{r}]_i\}$$

*as the space of learnable distributions for a Markovian and path-dependent parameterizations. Under these conditions, $\dim \text{span}(\mathcal{W}_M) \leq \dim \text{span}(\mathcal{W}_P)$ almost surely (with respect to $\mathbf{b}$) for all state-embedding functions $\psi \colon \mathcal{P} \to \mathbb{R}^d$, and $\dim \text{span}(\mathcal{W}_M) < \dim \text{span}(\mathcal{W}_P)$ for some $\psi$, including the natural embedding $\psi(p_i) = i$.*

In Proposition 3.1, the span (span) and dimension (dim) are meant in the Euclidean sense. In a nutshell, it states that an MDP augmented (or lifted) with a specific linear deterministic dynamics posseses significantly greater representational power than a simple linear MDP.

**Empirical illustration.** To illustrate Proposition 3.1, we consider a distribution $R \colon \mathcal{Q} \to \mathbb{R}_+$ with the form

$$R(q_i) = \sum_{1 \leq j \leq 4} w_j \exp\{-|i - k_j|\}, \quad (5)$$

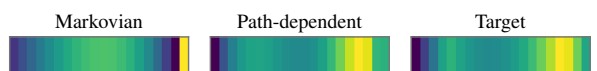

*Figure 3.* **Illustration of Proposition 3.1.** While a linear Markovian model (left) completely misses the high-probability regions of the target (right; see Equation (5)), our path-dependent parameterization (middle) provides a near-perfect distributional fit.

in which $k_j \in \{1, \ldots, N\}$ for $1 \le j \le 4$ and $w_j \in [0, 1]$ with $\sum_{j=1}^{4} w_j = 1$ as weighting factors; see Figure 3 (right). We also let $\psi$ be $\mathcal{P}$'s natural embedding, i.e., $\psi(p_i) = i$ for each $1 \le i \le N$. For simplicity, we let $N = 16$ and $M = 1$, and train both models by minimizing the TB loss; see Section 4 and Corollary B.1 for further details. As expected, Figure 3 shows that the distribution learned by our non-Markovian model closely matches the target distribution, while its Markovian alternative catastrophically underrepresents the target's high-probability regions.

### 3.2. GNN-based policies

Notably, the proven additional expressivity of path-dependent models can also be extended beyond linear policies. Drawing on the recent analysis in (Silva et al., 2025a; Kim et al., 2025a) and on the long-standing theory about the limitations of graph neural networks (GNNs; Kipf & Welling, 2017; Hamilton, 2020; Bronstein et al., 2021; Corso et al., 2024), we will demonstrate that the lifting process illustrated in Figure 1b provably boosts the expressive power of GNN-parameterized policy networks for discrete amorized samplers. For this, we briefly review the setup in (Silva et al., 2025a) and the structure of GNNs (Xu et al., 2020; Wang et al., 2022) for learning on dynamic graphs.

To start with, the reader should recall that a GNN modifies a featured graph $\mathcal{G} = (\mathbf{X}, \mathcal{E})$ with node features $\mathbf{X} \in \mathbb{R}^{n \times d}$ and edges $\mathcal{E}$ via a neighborhood-aggregating transformation,

$$\mathbf{x}'_i = \mu\left(\{\zeta(\mathbf{x}_j) \colon (i, j) \in \mathcal{E}\}\right) \coloneqq f(\mathbf{x}_i; \mathcal{G}),$$

in which $\mathbf{x}_i \in \mathbb{R}^d$ is the feature of the $i$-th node; $\zeta \colon \mathbb{R}^d \to \mathbb{R}^{d'}$, a parametric function (e.g., an MLP); and $\mu$, a permutation-invariant function on $\mathbb{R}^{d'}$ (Hamilton, 2020). Based on this formulation, it is clear that a single-layer GNN will fail to distinguish nodes $\mathbf{x}_i$ and $\mathbf{x}_j$ for which $\mathbf{x}_i = \mathbf{x}_j$ and both $i$ and $j$ have isomorphic 1-hop neighborhoods. In fact, the expressivity of deep GNNs is well-understood under the light of the Weisfeiler-Lehman hierarchy (WL; Weisfeiler & Lehman, 1968; Grohe, 2022).

Building on this, Silva et al. (2025a) have shown that a GFlowNet constructed on a simple edge-adding generative process (see Figure 4) with a policy function defined by

$$p_F(\mathcal{G}^{+(m,n)}|\mathcal{G}) \propto \exp\{\zeta(f(\mathbf{x}_m; \mathcal{G}) \oplus f(\mathbf{x}_n; \mathcal{G}))\}, \quad (6)$$

in which $\mathcal{G}^{+(m,n)}$ is the graph resulting from adding the

edge $(i, j)$ to $\mathcal{G}$, $\oplus$ is the concatenation operator, $f$ is a 1-WL multi-layer GNN, and $\zeta \colon \mathbb{R}^{2d'} \to \mathbb{R}$ is a feedforward deep network, cannot sample from some simple graph-structured distributions. This is another manifestation of state aliasing, as the sampler fails to distinguish distinct states due to its limited representational capacity. We show below that our path-dependent parameterization mitigates this issue.

**Proposition 3.2** (Temporal GNNs improve GFlowNet's expressivity)**.** *Let* SGs *be the space of state graphs defined by the edge-additive process over featured graphs (Figure 4), and let* $\mathcal{R}(\mathcal{S})$ *be the space of* all *unnormalized distributions over the graphs in* $\mathcal{S} \in$ SGs*. Also, by letting* $\mathbf{A}_i$ *be the adjacency matrix of the $i$-th graph and* $\mathbf{1} \in \mathbb{R}^{|\mathbf{X}|}$ *be column vector of 1's, define*

$$\mathbf{m}^{(o)} = \mathbf{0}, \ \mathbf{m}^{(i+1)} = \mathbf{m}^{(i)} + \alpha(\mathcal{G}_i) \cdot (\mathbf{A}_{i+1} - \mathbf{A}_i)\mathbf{1}$$

*and* $\mathbf{V}^{(i+1)} = [\mathbf{X}||\mathbf{m}^{(i+1)}] \in \mathbb{R}^{n \times (d+1)}$ *and* $\alpha(\mathcal{G}_i) \in (0, 1)$*. Similarly, for* $\mathcal{S} \in$ SGs*, let* $\mathcal{R}_P(\mathcal{S}) \subseteq \mathcal{R}(\mathcal{S})$ *be the set of distributions that a path-dependent amortized sampler based on a GNN $f_P$ and a feedforward deep network $\zeta_P$,*

$$p_F(\mathcal{G}^{+(m,n)}|\mathcal{G}_i) \propto \exp\{\zeta_P(f_P(\mathbf{v}_m^{(i)}; \mathcal{G}) \oplus f_P(\mathbf{v}_n^{(i)}; \mathcal{G}))\}, \quad (7)$$

*can sample from, and let* $\mathcal{R}_M(\mathcal{S}) \subseteq \mathcal{R}(\mathcal{S})$ *be the set of distributions learnable by an amortized sampler in the form of Equation (6). Then,* $\mathcal{R}_M(\mathcal{S}) \subseteq \mathcal{R}_P(\mathcal{S})$ *for every* $\mathcal{S} \in$ SGs*, and there exist sets $S$ for which* $\mathcal{R}_M(\mathcal{S}) \neq \mathcal{R}_P(\mathcal{S})$*.*

Intuitively, Proposition 3.2 states that including the information of when a node's degree has changed (Equation (7)) into the generative process strictly improves the expressivity of the learned sampler when compared against the original formulation in Equation (6). In doing so, the resulting generative process clearly fits into the path-dependent framework described in Figure 1b, as the policy function no longer satisfies the Markovian property. With this in mind, it would be interesting to understand the performance and limitations of this family of models for graph-structured tasks (e.g., Pandey et al. (2024)) under the lens of the analysis in (Souza et al., 2022). We leave this investigation to future endeavors.

**Empirical illustration.** To shed light on Proposition 3.2, we present in Figure 4a a state graph $\mathcal{S}$ for which $\mathcal{R}_P(\mathcal{S}) \neq \mathcal{R}_M(\mathcal{S})$. Notice, in particular, that $\mathcal{S}$ has two terminal nodes ($q_1$ and $q_2$) preceded by a non-terminal node ($p$). Due to the indistinguishability of node pairs $(a, b)$ and $(a, c)$ in $p$, a GFlowNet based on Equation (6) will fail to learn any non-uniform distribution over $q_1$ and $q_2$. In contrast, as $a$ and $c$ are modified at distinct moments in the different trajectories leading to $p$, our path-dependent parameterization in Equation (7) is able to fit any distribution. This fact is illustrated in Figure 4b, wherein we let

$$R(q_1) = \alpha \text{ and } R(q_2) = 1 - \alpha$$

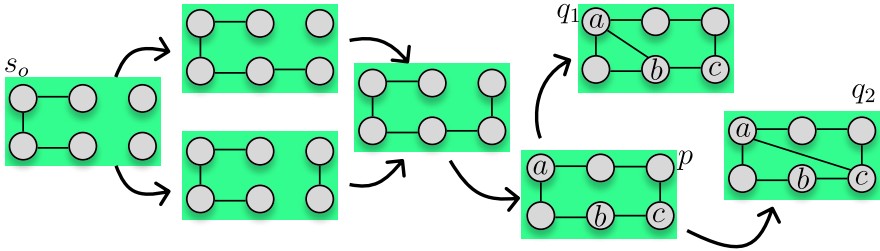
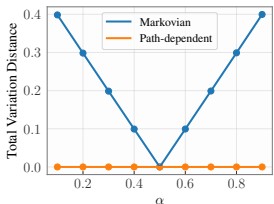

*(a)* Common GNN architectures (e.g., GCN, GAT) cannot distinguish the edges $(a, b)$ and $(a, c)$ on the graph $p$. Consequently, a GFlowNet parameterized as in Equation (6) would fail to learn any non-uniform distribution over $\{q_1, q_2\}$ when trained on this state graph.

*(b)* However, due to its larger expressive power, a path-dependent sampler perfectly fits any target distribution.

*Figure 4.* **Illustration of Proposition 3.2.** We present an example in which a 1-WL GNN-based Markovian amortized sampler (Equation (6)) cannot sample from an non-uniform target over $\{q_1, q_2\}$, while a corresponding path-dependent model can (right).

for $\alpha \in \{0.1, 0.2, \ldots, 0.9\}$ and use a standard two-layer GCN (Kipf & Welling, 2017) for parameterizing the policies, which are trained by minimizing the TB loss in Corollary B.1. As expected, the total variation (TV) distance between the target $R$ and the GFlowNet's learned distribution, which is uniform (as discussed above), is

$$\mathrm{TV}(R, p_{\mathsf{T}}) := \frac{1}{2} \left( \left| \alpha - \frac{1}{2} \right| + \left| (1 - \alpha) - \frac{1}{2} \right| \right) = \left| \alpha - \frac{1}{2} \right|,$$

while the same TV distance for our path-dependent model in Equation (7) is vanishingly small.

### 3.3. Multi-layer ReLU networks

A ReLU network learns a representation of each state by composing piecewise-linear functions, $f_i \colon \mathbb{R}^{d_i} \to \mathbb{R}^{d_{i+1}}$, defined by $f_i(\mathbf{x}) = \max(\mathbf{W}_i \mathbf{x}, 0)$, in which the max operator is applied elementwise; $d_o$ is the dimension of each state's natural embedding into the Euclidean space (e.g., $d_o = 1$ for the LINES environment). Although the composition $f_1 \circ f_2$ can approximate any target continuous function from $\mathbb{R}^{d_o}$ to $\mathbb{R}^{d_2}$ given a sufficiently large $d_1$ (Hornik et al., 1989), finding the appropriate parameters $\mathbf{W}_o$, $\mathbf{W}_1$, and $\mathbf{W}_2$ by gradient descent is often difficult—especially when the target has a large Lipschitz constant (Scaman & Virmaux, 2019). In the realm of RL, this limitation commonly manifests itself in the form of *state aliasing*, in which the learned model is unable to accurately distinguish states $\mathbf{x}_1$ and $\mathbf{x}_2$, regardless of their corresponding optimal action distributions. In other words, from the agent's perspective, the environment becomes *partially observable*.

To understand how this phenomenon emerges in the context of Markovian amortized samplers, consider the LINES environment with $M = 2$ and the target $R$ over $\mathcal{Q}$ depicted in Figure 5a. There, $R(q_2) = 1000 R(q_1)$ and $R(q_{20}) = R(q_2)$. Under these conditions, the optimal policy $p_F^{\star}$ at $p_2$ and $p_3$ differ significantly. Hence, as shown in Figure 5b, a Markovian sampler based on a 2-layer ReLU network with a softmax output cannot easily separate $p_2$ and $p_3$, as the

Kullback-Leibler divergence (Kullback & Leibler, 1951)

$$\max_{\tau \colon p_o \rightsquigarrow p_2, \tau' \colon p_o \rightsquigarrow p_3} \mathrm{KL} \left[ p_F(\cdot | p_3, \tau') || p_F(\cdot | p_2, \tau) \right] \quad (8)$$

remains essentially constant and near 0 throughout training (Figure 5b, blue). Contrarily, a path-dependent model—which has the ability to learn distinct values of $p_F(\cdot | p_2, \tau)$ for distinct $\tau$—can smoothly learn a distinct representation for $p_2$ and $p_3$ early in training (Figure 5b, orange). This also results in faster learning convergence and a better fit to the target distribution given a limited budget of trajectories; see Figure 5c, in which we used the model in Equation (4) for modelling path-dependence.

## 4. Path-dependent Amortized Samplers

The last section showed that a path-dependent algorithm can provide a provable expressivity boost for discrete amortized samplers. Nonetheless, it left the following question unanswered: how to define the latent dynamical process $\{W_t\}_{t \geq 1}$? After presenting a formal definition of our framework (Section 4.1 and Definition 4.1), we introduce a general-purpose algorithm for learning the vector field $\phi$ characterizing the latent dynamical system $\{W_t\}_{t \geq 1}$— through the rule $W_t = W_{t-1} + \phi(W_{t-1}, s_t)$—and demonstrate its correctness (Section 4.2 and Proposition 4.5).

### 4.1. Path-dependent Discrete Amortized Inference

Based on the prior examples, we provide a formal description of our framework for path-dependent discrete amortized inference. Crucially, although Proposition 3.2 suggests that different applications may benefit from specialized implementations, we will see in Sections 4 and 5 that a single architecture inspired by recent advances in recurrent RL (Irie et al., 2022; Irie & Gershman, 2025) is able to outperform standard Markovian models on most benchmark problems. With this in mind, we start by defining a *lifted pointed DAG*.

**Definition 4.1** (Lifted Pointed DAG). Let $\mathrm{SG} = (\mathcal{S}, \mathcal{X}, E)$ be a pointed DAG (as per Definition 2.1) with state space $\mathcal{S}$,

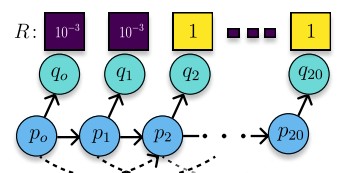
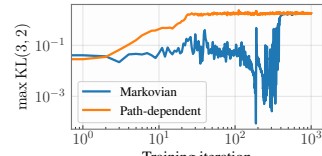
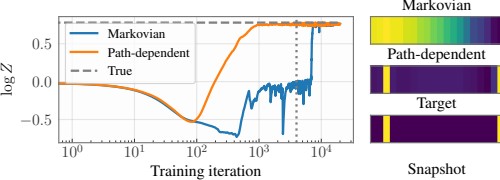

*(a)* Sparse LINES environment; $R(q_i) = 10^{-3}$ for all $i \notin \{2, 20\}$.

*(b)* KL-divergence between policies at $p_3$ and $p_2$; see Equation (8).

*(c)* Learning convergence for a Markovian and a path-dependent sampler for the sparse LINES environment.

*Figure 5.* **Markovian samplers struggle to distinguish similar states** (Section 3.3). (a) LINES environment with a sparse target; the optimal action distribution at $p_2$ puts a large weight on moving up (due to the largeness of $R(q_2)$), while the one at its adjacent state, $p_3$, is biased towards moving to the right. (b) KL-divergence between the policies at $p_2$ and $p_3$ throughout training; a ReLU network-parameterized Markovian sampler takes around 100 times more iterations to separate $p_2$ and $p_3$ than its path-dependent counterpart. (c) This results in a better convergence rate. The right-hand-side snapshot shows the sampler's state at the iteration marked by the dotted line.

edges $E$, and terminal states $\mathcal{X}$. Also, let $\phi\colon \mathcal{S} \times \Omega \to \Omega$ be a function over some set $\Omega$, and $W_o \in \Omega$. We say that $\mathrm{SG}^\star = (\mathcal{S}^\star, \mathcal{X}, E^\star, \phi, W_o)$ is a *pointed DAG lifted by $\phi$ and $W_o$* when the following conditions are satisfied.

1. $\mathcal{S}^\star \subseteq \mathcal{S} \times \Omega$, $E^\star \subseteq \mathcal{S}^\star \times (\mathcal{S}^\star \cup \mathcal{X})$, $s_o^\star := (s_o, W_o) \in \mathcal{S}^\star$;

2. $(s, W) \in \mathcal{S}^\star$ iff there is a path $\{(s_t, W_t)\}_{t=0}^T$ such that $(s_T, W_T) = (s, W)$, $W_t = W_{t-1} + \phi(s_t, W_{t-1})$, and $(s_{t-1}, s_t) \in E$ for all $1 \le t \le T$;

3. $((s, W), (s', W')) \in E^\star$ iff $(s, s') \in E$ are connected in SG and $W' = W + \phi(s', W)$; and

4. $((s, W), x) \in E^\star$ iff $(s, W) \in \mathcal{S}^\star$ and $(s, x) \in E$.

When $\phi$, $W_o$, and SG are clear from the context, we shall refer to $\mathrm{SG}^\star$ simply as the lifted pointed DAG.

In Definition 4.1, we will often let $\Omega = \mathbb{R}^d$ for some integer $d$. As illustrated in Figure 1b, a lifted pointed DAG extends a state graph with a latent dynamical system $\{W_t\}_{t \ge 1}$ abiding by the difference equation determined by $\phi$ and conditioned on $\{s_t\}_{t \ge 1}$ (*state-conditional*). In the following examples, we show that the problems outlined in the previous section fit into the structure presented in Definition 4.1.

*Example* 4.2 (LINES). In the context of Figure 2, $\mathcal{S} = \mathcal{P}$, $\mathcal{X} = \mathcal{Q}$, and $E = \bigcup_{i=0}^N \{(p_i, p_{i+k}) \text{ for } 1 \le k \le \min\{M, N-i\}\} \cup \{(p_i, q_i)\}$. For Proposition 3.1, $W_o = \mathbf{a}_o$, $\Omega = \mathbb{R}^d$ ($\mathbf{b}$ is fixed), and $\phi\colon \mathcal{S} \times \mathbb{R}^d \to \mathbb{R}^d$ is defined by

$$\phi(s, \mathbf{a}) = (\mathbf{w}_a^T \psi(s)) \cdot \mathbf{v}_a,$$

in which $\mathbf{w}_a$ and $\mathbf{v}_a$ are learnable parameters and $\psi$ is an embedding function. Also, notice that $\phi$ is independent of $\mathbf{a}$.

*Example* 4.3 (GRAPHS). We consider the edge-additive process illustrated in Proposition 3.2. There, $\mathcal{S}$ and $\mathcal{X}$ are sets of featured graphs with a fixed number $N$ of nodes, denoted by $\mathcal{G} = (\mathbf{X}, \mathcal{E})$; $\mathcal{E}$ is the set of edges. Also, $E = \{(\mathcal{G}, \mathcal{G}') \text{ for } |\mathcal{E}' \setminus \mathcal{E}| = 1 \text{ and } \mathcal{E} \subseteq \mathcal{E}'\}$. For instance, $|\mathcal{X}| = 2$ and $|\mathcal{S}| = 5$ in Figure 4a. In Proposition 3.2, $W_o =$

$(\mathbf{m}_o, \mathbf{A}_o)$, with $\mathbf{m}_o = \mathbf{0} \in \mathbb{R}^N$ and $\mathbf{A}_o$ as the initial state's adjacency matrix, $\Omega = \mathbb{R}^N \times \{1, 0\}^{N \times N}$, and $\phi\colon \mathcal{S} \times \Omega \to \Omega$ is characterized by the recurrence relation

$$\phi(s, (\mathbf{m}, \mathbf{A})) = (\alpha(s) \cdot (\mathbf{A}_s - \mathbf{A})\mathbf{1}, \ \mathbf{0}_{N \times N}),$$

in which $\alpha\colon \mathcal{S} \to (0, 1]$, $\mathbf{1}$ is a $N$-sized vector of 1s, and $\mathbf{A}_s$ is $s$'s adjacency matrix. Correspondingly, the update step for the dynamic system $\{W_t\}_{t \ge 1}$ can be written as

$$\mathbf{A}_{t+1} = \mathbf{A}_t, \ \mathbf{m}_{t+1} = \mathbf{m}_t + \alpha_t (\mathbf{A}_{t+1} - \mathbf{A}_t) \mathbf{1},$$

with $\alpha_t = \alpha(s_t)$. Here, $\phi$ depends on $s$ and $W = (\mathbf{m}, \mathbf{A})$.

Interestingly, a stochastic latent dynamical system can also be developed (e.g., as in Särkkä & Solin (2019)). Nonetheless, this significantly increases the problem's complexity, as the lifted domain becomes uncountable; see Section D for a formal exposition and preliminary derivations.

### 4.2. Choosing a latent dynamical system

**Recurrent policy network.** To select an appropriate update rule for $\{W_t\}_{t \ge 1}$, we initially remark upon the fact that

$$W_t = W_{t-1} + \phi(s_t, W_{t-1})$$

can be directly implemented by a recurrent neural network (RNN; Rumelhart et al., 1986; Cho et al., 2014). A single-layer vanilla RNN with identity activation, for instance, would provide the equation

$$\phi(s_t, W_t) = \psi(s_t)\mathbf{W}_s + W_{t-1}(\mathbf{W}_h - \mathbf{I}) + \mathbf{b}_h,$$

with $W_t \in \mathbb{R}^d$, $\mathbf{W}_s \in \mathbb{R}^{d \times d}$, $\mathbf{W}_h \in \mathbb{R}^{d \times d}$, and $\mathbf{b}_h \in \mathbb{R}^d$ as optimizable parameters and $\psi\colon \mathcal{S} \to \mathbb{R}^d$ as a (possibly learnable) embedding function for $\mathcal{S}$. The resulting representation, $W_t$, could then be used as an input to a standard multi-layer neural network for computing a probability distribution over $s_t$'s children in the state graph (as in Equation (3)). In practice, this process would be repeated until the sampling of a terminal state $x \in \mathcal{X}$, as explained in Section 2. We refer to a forward policy $p_F$ parameterized by such a recurrent network as a *recurrent policy network* (Karkus et al., 2017; Ni et al., 2022).

**Self-Referential Weight Matrix (SRWM).** Based on the long-standing success of Fast Weight Programmers (FWPs; Schlag et al., 2021; Irie & Gershman, 2025) on RL and on our own empirical assessment (see Section 5), we propose using a modified self-referential weight matrix (SRWM; Schmidhuber, 1993b; Irie et al., 2022) for learning the sampler's recurrent policy network. Simply put, our modified SRWM implements an update rule with the form:

$$\mathbf{q}_t, \mathbf{k}_t, \mathbf{v}_t, \beta_t = \mathbf{W}\psi(s_t) + \mathbf{b},$$
$$\bar{\mathbf{v}}_t = \mathbf{W}_{t-1}\zeta(\mathbf{k}_t),$$
$$\mathbf{W}_t = \mathbf{W}_{t-1}\mathbf{R} + \sigma(\beta_t) \cdot (\mathbf{v}_t - \bar{\mathbf{v}}_t) \otimes \zeta(\mathbf{q}_t),$$
(9)

in which $\mathbf{W} \in \mathbb{R}^{(3d+1)\times d}$ and $\mathbf{b} \in \mathbb{R}^{3d+1}$ are parameters, $\zeta \colon \mathbb{R}^d \to \Delta_{d-1}$ is an activation function that maps a vector in $\mathbb{R}^d$ into the $(d-1)$-dimensional simplex (e.g., softmax), $\mathbf{R}$ is a (fixed) ergodic rotation matrix, and $\sigma(x) = (1+\exp\{-x\})^{-1}$ is the sigmoid function. Intuitively, $\mathbf{R}$ enforces diversity into the policy by rotating the eigenvectors of $\mathbf{W}_t$ throughout the rollout of a trajectory, which reduces the aliasing-related effects described in Section 3.3. Under the terminology introduced by Schmidhuber (1993b;a), $\mathbf{W}$ and $\mathbf{b}$ are *slowly changing* variables optimized via stochastic gradient descent, while $\mathbf{W}_{t-1}$ is a *rapidly evolving* matrix with a self-determined learning rate $\sigma(\beta_t)$. Correspondingly, we also parameterize the initial state for the latent dynamical system as $\mathbf{W}_o = \mathbf{a}_o\mathbf{b}_o^T$, in which both $\mathbf{a}_o \in \mathbb{R}^{d\times 1}$ and $\mathbf{b}_o \in \mathbb{R}^{d\times 1}$ are optimizable parameters.

Importantly, we notice that other architectures, such as LSTMs (Hochreiter & Schmidhuber, 1997) and GRUs (Cho et al., 2014), have also been tested for parameterizing the update rule $\phi$ in early experiments. However, we found them to be generally less effective than a comparably sized SRWM, which corroborates evidence in prior work (Irie et al., 2022).

### 4.3. Learning a latent dynamical system

**Criteria for path-dependent samplers.** Given the SRWM-based architecture, we are left with the problem of estimating the model's parameters. In this context, we demonstrate that the learning criteria for GFlowNets can also be used for learning a policy function on our lifted DAG. (Interested readers are referred to Section B for an extension of the flow network analogy to our context). We start by recalling the definition of a *state flow* (Bengio et al., 2021).

**Definition 4.4.** A *state flow* for a target $R\colon \mathcal{X} \to \mathbb{R}_+$ is any function $F\colon \mathcal{S}^\star \cup \mathcal{X} \to \mathbb{R}_+$ s.t. $F(x) = R(x)$ for all $x \in \mathcal{X}$.

In practice, $F$ is a learned function representing the total flow arriving at each $s^\star \in \mathcal{S}^\star$. Under certain conditions (established below), this is also equivalent to the unnormalized marginal probability of reaching $s^\star$ when starting at the initial state $s_o^\star$ and following the forward policty $p_F$ (Bengio et al., 2021). As such, it is natural to restrict $F$ in $\mathcal{X}$

by letting $F(x) = R(x)$ for $x \in \mathcal{X}$ (Deleu et al., 2022), as we do in Definition 4.4. From this perspective, we show in the next proposition that Madan et al. (2022)'s subtrajectory balance (SubTB) condition is enough to ensure sampling correctness on the lifted DAG of Definition 4.1.

**Proposition 4.5.** *Let $\mathcal{S}^\star$ be a lifted pointed DAG, and $p_F$, $p_B$, and $F$ be a recurrent forward policy, a backward policy, and a flow function on $\mathcal{S}^\star$. Assume that, for each complete trajectory $\tau = (s_t^\star)_{t=0}^T$ with $s_T^\star = x \in \mathcal{X}$ and each $i < j$,*

$$F(s_i^\star)p_F(s_{i+1:j}^\star|s_i^\star) = p_B(s_{i:j-1}^\star|s_j^\star)F(s_j^\star), \quad (10)$$

*in which $p_F$ is computed according to the graphical model in Figure 1b and $p_B(s_i^\star|s_{i+1:j}^\star) = \prod_{t=i}^{j-1} p_B(s_t^\star|s_{t+1}^\star)$, according to the transposed SG$^\star$. Then, denoting by $\tau \rightsquigarrow s^\star$ the trajectories starting at $s_o^\star$ and finishing at $s^\star \in \mathcal{S}^\star \cup \mathcal{X}$,*

$$p_\mathsf{T}(s^\star) = \sum_{\tau \rightsquigarrow s^\star} p_F(\tau|s_o^\star) \propto F(s^\star). \quad (11)$$

*In particular, $p_\mathsf{T}(x) \propto R(x)$ for $x \in \mathcal{X}$.*

Since our interest lies exclusively on the sampling of complete objects, it should be clear that the Equation (11) remains true on $\mathcal{X}$ when Equation (10) is only for $i = 0$ and $j = T$ (Madan et al., 2022). This results in the trajectory balance (TB) condition (Malkin et al., 2022).

**Corollary 4.6.** *Similarly to Proposition 4.5, assume that*

$$F(s_o^\star)p_F(\tau|s_o^\star) = R(x)p_B(\tau|x) \quad (12)$$

*for each complete trajectory $\tau = (s_t^\star)_{t=0}^T$. Then, $p_\mathsf{T}(x) \propto R(x)$ for each terminal state $x \in \mathcal{X}$.*

As in Bengio et al. (2023, Example 6), Corollary 4.6 only depends on the flow associated to the initial state, $F(s_o^\star)$, which equals the target $R$'s partition function when Equation (12) is satisfied (Malkin et al., 2022). Below, we demonstrate that the $F$-independent contrastive balance (CB) condition (Zhang et al., 2023b; da Silva et al., 2024a) is also sufficient to ensure the marginal of $p_F$ over $\mathcal{X}$ matches $R$.

**Corollary 4.7.** *Under the setup of Proposition 4.5, let $\tau = (s_t^\star)_{t=0}^T$, $\tau' = (s_t'^\star)_{t=0}^T$ be complete trajectories on SG$^\star$. If*

$$p_F(\tau|s_o^\star)p_B(\tau'|s_T'^\star)R(s_T'^\star) = p_F(\tau'|s_o^\star)p_B(\tau|s_T^\star)R(s_T^\star)$$

*for every pair $(\tau, \tau')$, then $p_\mathsf{T}(x) \propto R(x)$.*

Taken together, Proposition 4.5 and Corolaries 4.6 and 4.7 demonstrate that traditional learning objectives for GFlowNets can be seamlessly adapted to a lifted pointed DAG (see Definition 4.1). In practice, we enforce these conditions by minimizing an expectation of the log-squared difference between their left- and right-hand sides (as in Bengio et al., 2023, Examples 4-6). Remarkably, when $p_B$ does not depend on the latent dynamical system $\{W_t\}_{t\geq 1}$, the optimal $p_F$ satisfying either Corollaries 4.6 or 4.7 is Markovian with respect to the unlifted DAG; see Proposition 4.8.

**Proposition 4.8.** *Under the setup of Proposition 4.5, assume $\tilde{p}_B$ is a backward policy on SG such that $p_B(s_t^\star|s_{t+1}^\star) = \tilde{p}_B(s_t|s_{t+1})$ for each trajectory $(s_t^\star)_{t=0}^T$ on $\mathrm{SG}^\star$. We write $s_t$ for the unlifted instantiation of $s_t^\star := (s_t, W_t)$. Let $p_F$ be a forward policy abiding by either Corollaries 4.6 or 4.7. Then, there is a policy $\tilde{p}_F$ on SG for which $p_F(s_{t+1}^\star|s_t^\star) = \tilde{p}_F(s_{t+1}|s_t)$.*

Figure 6 illustrates Proposition 4.8 for the LINES environment (recall Figure 2), measuring the same-state, distinct-trajectory KL divergence averaged over non-terminal states $s \in \{p_1, \ldots, p_N\}$ when (i) $p_B$ is Markovian and (ii) $p_B$ is non-Markovian. Under (i), $p_F$ converges to a Markovian policy, as indicated by the vanishing KL divergence; under (ii), $p_F$ stabilizes as a non-Markovian model.

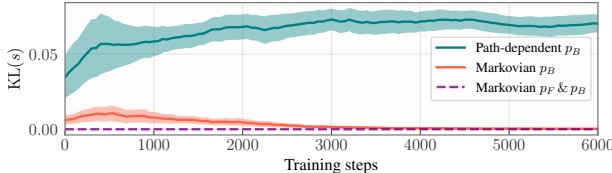

*Figure 6.* **Illustration of Proposition 4.8**. A (non-)Markovian $p_B$ entails a (non-)Markovian $p_F$ at equilibrium. We let $\mathrm{KL}(s) = \max_{\tau,\tau': s_o \rightsquigarrow s} \mathrm{KL}(p_F(\cdot|\tau,s)\|p_F(\cdot|\tau',s))$ be the maximum KL between policies conditioned on the same state, but distinct trajectories; $\mathrm{KL}(s) > 0$ for some $s$ indicates $p_F$ is path-dependent.

Importantly, in view of the aliasing-related challenges described in Section 3, the choice of a Markovian $p_B$ can be effective, as a path-dependent parameterization may provide a better approximation to the optimal policy $\tilde{p}_F$ outlined in Proposition 4.8 than its Markovian equivalent.

## 5. Experiments

The main purpose of our experiments is to attest the effectiveness of our path-dependent models in terms of their goodness-of-fit to the target distribution relatively to their Markovian counterparts. For this, we consider the following four standard benchmark tasks in the GFlowNet literature. Additionally, we provide results for the tasks of preference learning, bit sequences, and lazy random walk in the supplement. We implemented our models in JAX (Bradbury et al., 2018); see Section F for further details.

**Set generation (Bengio et al., 2023).** This task consists of generating fixed-size subsets of a fixed superset. Let $\Gamma = \{1, 2, \ldots, S\}$ be such a superset, and $u\colon \Gamma \to \mathbb{R}$ be a log-utility function for the elements of $\Gamma$. In this context, our initial state is the empty set, $s_o = \emptyset$, and each transition consists of adding an element of $\Gamma \setminus s$ to the current state $s$ until $|s| = K$. Our objective is to sample each $\gamma \subseteq \Gamma$ with $|\gamma| = K$ in proportion to $R(\gamma) = \sum_{i \in \gamma} \exp\{u(i)\}$. In this scenario, Table 1 shows that a path-dependent

*Table 1.* FCS for set generation with $S = 64$ and varying $K$.

| $K$ | Markovian | Path-dependent |
|---|---|---|
| 16 | $0.091\pm_{0.012}$ | $\mathbf{0.053}\pm_{0.006}$ |
| 24 | $0.105\pm_{0.009}$ | $\mathbf{0.064}\pm_{0.002}$ |

*Table 2.* FCS for the sequence design task. Syn-$S$ stands for the synthetic distribution over $S$-sized sequences, while SIX6 and PHO4 refer to the task of DNA sequence generation.

| Task | Markovian | Path-dependent |
|---|---|---|
| Syn-8 | $0.084\pm_{0.001}$ | $\mathbf{0.011}\pm_{0.001}$ |
| Syn-16 | $0.165\pm_{0.006}$ | $\mathbf{0.024}\pm_{0.001}$ |
| Syn-32 | $0.243\pm_{0.015}$ | $\mathbf{0.055}\pm_{0.002}$ |
| SIX6 | $0.165\pm_{0.006}$ | $\mathbf{0.135}\pm_{0.003}$ |
| PHO4 | $0.175\pm_{0.004}$ | $\mathbf{0.130}\pm_{0.001}$ |

parameterization achieves better goodness-of-fit for $S = 64$ and $K \in \{16, 32\}$ ($\mathcal{X}$ contains roughly $10^{14}$ and $10^{18}$ elements, respectively) in terms of FCS (i.e., the average TV distance on tractable subsets of $\mathcal{X}$; see Equation (18)).

**Sequence design (Jain et al., 2022).** For this problem, we autoregressively generate sequences with a fixed size $S$. That is, we start at an empty sequence $s_o = \emptyset$ and add tokens from a given vocabulary $\mathcal{V}$ to the current state $s$ until $s \in \mathcal{V}^S$. First, we consider a synthetic distribution with $\mathcal{V} = \{1, \ldots, 6\}$, $S \in \{8, 16, 32\}$, and $R(s) = \sum_{i=1}^S u(i) \cdot v(s_i)$ with $u\colon \{1, \ldots, S\} \to \mathbb{R}_+$ and $v\colon \mathcal{V} \to \mathbb{R}_+$ as utility functions. Table 2 shows that our path-dependent parameterization has a better fit to the target distribution for each $S$. Figure 10 (supplement) shows these results are consistent across model sizes. Second, and similarly, we show in Table 2 that our model also achieves better distributional fit for the task of 8 (resp. 10)-sized biological sequence design of Jain et al. (2022), in which the target distribution measures the binding affinity of a DNA sequence to human (resp. yeast) transcription factors (Barrera et al., 2016; Trabucco et al., 2022) denoted by SIX6 (resp. PHO4).

**Grid world (Bengio et al., 2021).** The grid world represents a sparse distribution over $\{0, \ldots, N-1\}^2$ illustrated in Figure 7 (right). The initial state is the origin, $s_o = (0,0)$, and a transition consists of either stopping, or moving up, or moving to the right, similarly to the LINES task in Figure 2. Again, we show in Figure 7 that our path-dependent model achieves a better approximation to the target than its Markovian counterpart—which struggles to navigate the space.



*Figure 7.* Goodness-of-fit to the target in the grid-world .

# 6. Conclusions

When working with continuous distributions, the most effective samplers rely on augmenting the state space with an auxiliary momentum (latent) variable to accelerate convergence (Neal et al., 2011; Cheng et al., 2018), a process which we call *lifting*. In this work, we demonstrated that a similar strategy can be used effectively for discrete amortized samplers in compositional spaces. We also delineated the limitations of existing samplers operating on the unlifted state space, and showed that the latent dynamical process can be efficiently learned with a self-referential weight matrix.

In closing, we note the question regarding the feasibility of a stochastic dynamics for a path-dependent model remains open, which we believe represents an interesting venue for future research. All in all, our results further suggest that algorithms for partially observable MDPs are well-suited for discrete amortized sampling, corroborating early observations by Randlov (1998); Boussif et al. (2024).

# Impact statement

Our work delineates the limitations of Markovian amortized samplers for discrete and compositional objects, which have attracted increasing interest from our community, and introduces a general-purpose path-dependent algorithm for mitigating them. However, we do not foresee any direct negative societal impacts for this work specifically.

# Acknowledgements

ESW acknowledges support from the CIFAR Learning in Machines and Brains program.

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

# A. Proofs

This section provides the proofs for our statements throughout the text. Each section corresponds to a different Proposition and, when needed, Corollary. Our results in Section 4.3 closely match that of (Bengio et al., 2023; Malkin et al., 2022; Madan et al., 2022), to which we refer the reader for further details.

## A.1. Proof of Proposition 3.1

Our proof has two steps. First, we characterize the space of learnable distributions for both the Markovian and path-dependent parameterizations. Second, we argue that, under the conditions of Proposition 3.1, the latter is larger than the former.

As $M = 1$, $\mathbf{W}_i = \mathbf{W} \in \mathbb{R}^{2 \times d}$ for the Markovian model. For clarity, let

$$\mathrm{Softmax}(\mathbf{W}\psi(p_i)) = \begin{bmatrix} \alpha_i & 1 - \alpha_i \end{bmatrix};$$

equivalently, for $\mathbf{y}_i = \mathbf{W}\psi(p_i) \in \mathbb{R}^2$, $\log \frac{\alpha_i}{1-\alpha_i} = \mathbf{y}_i^T(\mathbf{e}_1 - \mathbf{e}_2)$, with $\mathbf{e}_1$, $\mathbf{e}_2 \in \{1, 0\}^{2 \times 1}$ as the first and second columns of the identity matrix. (As per standard convention, vectors in $\mathbb{R}^d$ represent column vectors). We omit $\mathbf{m}_i$ as $\mathbf{m}_i = \mathbf{1}$ for each $i$ except for $i = N$—in which case $\alpha_N = 0$ is the only solution.

From this viewpoint, the sampler's objective can be represented through the linear system

$$\mathbf{o} = \Psi \mathbf{k}, \tag{13}$$

in which $\mathbf{o} \in \mathbb{R}^N$, $\mathbf{o}_i = \log \alpha_i/1-\alpha_i$, and $\Psi$, $\Psi_i = \psi(p_i)^T$, are fixed[1]; $\mathbf{k}_i = \mathbf{W}^T(\mathbf{e}_1 - \mathbf{e}_2)$ is the quantity we wish to learn (through $\mathbf{W}$). Consequently, we can only learn distributions for which the policies' log-odds, $\mathbf{o}$, are within the column space of the state-embedding matrix, $\Psi$. Under the notation of Proposition 3.1, this implies that $\dim \mathcal{W}_M = \mathrm{rank}\ \Psi \leq d$.

Under the non-Markovian parameterization of our linear control system, the corresponding learning problem becomes

$$\mathbf{o}_i = \psi(p_i)^T \mathbf{a}_i \mathbf{b}^T (\mathbf{e}_1 - \mathbf{e}_2),$$

in which

$$\mathbf{a}_i = \mathbf{a}_o + \left[ \sum_{0 \leq j \leq i} \mathbf{w}_a^T \psi(p_i) \right] \mathbf{v}_a.$$

Clearly, $\delta := \mathbf{b}^T(\mathbf{e}_1 - \mathbf{e}_2)$ is a non-zero scalar almost surely (with respect to $\mathbf{b}$). Therefore, we let $\tilde{\mathbf{o}}_i = \mathbf{o}_i/\delta$. Similarly, define

$$\mathbf{c}_i = \sum_{0 \leq j \leq i} \psi(p_i) \tag{14}$$

as the cumulative sum of the state-embeddings $\psi(p_j)$. Then, $\mathbf{a}_i = \mathbf{a}_o + (\mathbf{w}_a^T \mathbf{c}_i)\mathbf{v}_a$, and

$$\tilde{\mathbf{o}}_i = \psi(p_i)^T \mathbf{a}_o + \psi(p_i)^T (\mathbf{w}_a^T \mathbf{c}_i)\mathbf{v}_a = \psi(p_i)^T \mathbf{a}_o + (\psi(p_i)^T \mathbf{v}_a) \cdot (\mathbf{c}_i^T \mathbf{w}_a).$$

Let $\mathbf{C}$ be the $N \times d$ matrix whose $i$th row is $\mathbf{c}_i$. From the equation above,

$$\tilde{\mathbf{o}} = \Psi \mathbf{a}_o + [\Psi \mathbf{v}_a] \odot [\mathbf{C}\mathbf{w}_a]. \tag{15}$$

As a consequence, our path-dependent linear model can represent any distribution for which the log-odds of the corresponding policies are within the linear space spanned by $\Psi$ and the Hadamard-span of $\mathbf{C}$ and $\Psi$. In other words, let

$$\mathcal{C} = \{\mathbf{v} \colon \mathbf{v} = \Psi \mathbf{w} \text{ for some } \mathbf{w}\} \text{ and } \mathcal{H} = \{\mathbf{v} \colon \mathbf{v} = (\mathbf{C}\mathbf{a}) \odot (\Psi \mathbf{b}) \text{ for some } \mathbf{a}, \mathbf{b}\}.$$

In this scenario, the dimension of the range of distributions representable by a Markovian parameterization is $\dim \mathcal{C}$, while a path-dependent model is capable of representing distributions in a space with dimension $\dim(\mathcal{C} + \mathcal{H})$. Hence, $\dim \mathcal{W}_P := \dim(\mathcal{C} + \mathcal{H}) \geq \dim \mathcal{C} =: \dim \mathcal{W}_M$ regardless of $\psi$, and it is clear that $\dim \mathcal{W}_P > \dim \mathcal{W}_M$ for some $\psi$; a trivial example is for $\psi(p_i) = \mathbf{1}_d \in \mathbb{R}^d$ constant, in which case $\mathcal{C} = \{\alpha \mathbf{1}_N \colon \alpha \in \mathbb{R}\} \subset \mathbb{R}^N$ and $\mathcal{H} = \{\alpha \begin{bmatrix} 0 & \dots & N-1 \end{bmatrix}^T \colon \alpha \in \mathbb{R}\}$; thus, $\dim \mathcal{C} = 1$ and $\dim(\mathcal{H} + \mathcal{C}) = 2$.

We found the case $d = 1$ with the natural embedding $\psi(p_i) = i$ to be instructive. Under these conditions, $\mathcal{C}$ is simply the spanned by $\mathbf{N} := \begin{bmatrix} 0 & 1 & \dots & N-1 \end{bmatrix}$, while $\mathcal{H}$ is the Hadamard product between $\mathbf{C} = \begin{bmatrix} c_o & c_1 & \dots & c_{N-1} \end{bmatrix}$, $c_i = \frac{i(i-1)}{2}$, and $\mathbf{N}$. Clearly, $\mathbf{N}$ and $\mathbf{C}$ are linearly independent for $N \geq 3$. The interested reader is invited to fill up the details.

---

[1] We omit the $N$th term from $\mathbf{o}$ as $\alpha_N = 0$ (the only possible move at $p_N$ is going to $q_N$).

## A.2. Proof of Proposition 3.2

Similarly to the prior section, this proof has two steps: we first demonstrate that each distribution representable by a Markovian parameterization can also be learned by its path-dependent counterpart; then, we show that there exist distributions that can only be learned by the latter.

On the one hand, it should be clear that the path-dependent model defined by the recurrent GNN is at least as expressive as its Markovian counterpart. To see this, simply notice that we can zero out the appended (right-most) coordinate of $\mathbf{V}$ in the first GNN layer; in doing so, the resulting model becomes equivalent to that of Equation (6). As a consequence, $\mathrm{R}_P(S) \subseteq \mathcal{R}_M(S)$.

On the other hand, consider the state graph $S^\star$ in Figure 4. As shown in the main text, a GFlowNet parameterized as in Equation (6) can only sample from an uniform distribution over the terminal states, $q_1$ and $q_2$. In contrast, as the edges $(a, b)$ and $(a, c)$ have distinct temporal features ($b$ is modified once during the generative process; $c$, twice), the actions resulting in $q_1$ and $q_2$ from $p$ are distinguishable by a recurrent GNN. Hence, our path-dependent model can approximate any distribution over $\{q_1, q_2\}$. That is, $\mathcal{R}_M(S^\star) \neq \mathcal{R}_P(S^\star)$. This concludes our demonstration.

## A.3. Proof of Proposition 4.5

We proceed with a direct computation of the marginal distribution, $p_{\mathsf{T}}$, of each augmented state $s^\star$ to show that

$$p_{\mathsf{T}}(s^\star) = \sum_{\tau \rightsquigarrow s^\star} p_F(\tau|s_o^\star) \propto F(s^\star);$$

as the space of terminal objects, $\mathcal{X}$, remains unchanged in the lifted DAG, this implies that the marginal distribution of our path-dependent model matches $R$ on $\mathcal{X}$ (up to a normalizing constant). It should be noted that the the number of trajectories for which $p_F(\tau|s_o^\star) > 0$ is finite in $\mathrm{SG}^\star$ due to the deterministic nature of the latent dynamical system. (The lack of finiteness is the reason for which designing a stochastic latent dynamical system is significantly harder than a deterministic one—see Section D).

Our asumption is that

$$F(s_i^\star)p_F(s_{i+1:j}^\star|s_i^\star) = p_B(s_{i:j-1}^\star|s_j^\star)F(s_j^\star)$$

for each slice of every trajectory $\tau = (s_t^\star)_{t=0}^T$. Fix $i = 0$. Then, the marginal distribution of $s^\star$ induced by $p_F(\cdot|s_o^\star)$ is

$$p_{\mathsf{T}}(s^\star) = \sum_{\tau:\, s_o^\star \rightsquigarrow s^\star} p_F(\tau|s_o^\star) = \sum_{\tau:\, s_o^\star \rightsquigarrow s^\star} p_B(\tau|s^\star) \cdot \frac{F(s^\star)}{F(s_o^\star)}.$$

Obviously, $\sum_{\tau:\, s_o^\star \to s^\star} p_F(\tau|s^\star) = 1$, as $s_o^\star$ is the only absorbing state for the Markov chain induced by the backward policy on $\mathrm{SG}^\star$. This can also be seen by a recursive characterization of this sum. Indeed, let $Q(s_o^\star) = 1$ and

$$Q(g^\star) = \sum_{\tau:\, s_o^\star \rightsquigarrow g^\star} p_B(\tau|g^\star)$$

for each state $g^\star$ on the lifted DAG. Then,

$$Q(s^\star) = \sum_{\tau:\, s_o^\star \rightsquigarrow s^\star} p_B(\tau|s^\star) = \sum_{g^\star \in \mathrm{Pa}(s^\star)} p_B(g^\star|s^\star) \sum_{\tau:\, s_o^\star \rightsquigarrow g^\star} p_B(\tau|g^\star) = \sum_{g^\star \in \mathrm{Pa}(s^{s^\star})} p_B(g^\star|s^\star)Q(g^\star).$$

Since $\sum_{g^\star \in \mathrm{Pa}(s^{s^\star})} p_B(g^\star|s^\star) = 1$, the above equation implies that $\mathbf{Q} := (Q(g^\star))_{g^\star \in \mathcal{S}^\star \cup \mathcal{X}}$ is the eigenvector of a stochastic matrix corresponding to the unit eigenvalue. By the finiteness of the DAG, $\mathbf{Q} = \mathbf{1}$ is the unique eigenvector (due to the boundary conditions, $Q(s_o^\star) = 1$, as $s_o^\star$ is an absorbing state for the backward policy). Formally, by writing $\mathbf{Q} = \mathbf{MQ}$ for a stochastic matrix $\mathbf{M}$, we may re-index the nodes of the underlying DAG in topological order. In doing so, $\mathbf{M}$ becomes upper-triangular, and the value of $\mathbf{Q}$ can be uniquely characterized via backward induction, given the boundary conditions on $s_o^\star$.

All in all, we observe that

$$p_{\mathsf{T}}(s^\star) = \frac{F(s^\star)}{F(s_o^\star)} \sum_{\tau:\, s_o^\star \rightsquigarrow s^\star} p_B(\tau|s^\star) \propto F(s^\star).$$

Corollaries 4.6, 4.7, B.1 follow directly from Proposition 3.2. In fact, Corollary 4.7 entails Corollary 4.6, and the global minima in Corollary B.1 satisfy either Corollary 4.6 or 4.7.

## A.4. Proof of Proposition 4.8

Before diving into a formal demonstration of the forward policy's path-independence, the reader should recall the graphical model in Figure 1b. If $p_B$ is independent of $\{W_t\}_{t \geq 1}$, so is $p_F$—as both induce the same distribution over trajectories when the (trajectory or contrastive) balance condition is satisfied. As a consequence, $p_F$ should also be independent of $\{W_t\}_{t \geq 1}$. The proof below builds on this intuition: we compute the joint distribution over trajectories induced by $p_F$ and convert it into a $p_B$-induced joint distribution through the TB condition. Then, we marginalize out all states following a given transition $(s_t, s_{t+1})$ to show that the probability of going from $s_t$ to $s_{t+1}$ does not depend on the states preceding $s_t$.

For conciseness, let $\bar{\mathcal{S}} = \mathcal{S} \cup \mathcal{X}$ and $\bar{\mathcal{S}}^{\star} = \mathcal{S}^{\star} \cup \mathcal{X}$. Let $p_B \colon \bar{\mathcal{S}}^{\star} \times \mathcal{S}^{\star} \to [0,1]$ and $\tilde{p}_B \colon \bar{\mathcal{S}} \times \mathcal{S} \to [0,1]$ be the backward policies from Proposition 4.8, which satisfy $p_B(s_t^{\star}|s_{t+1}^{\star}) = \tilde{p}_B(s_t|s_{t+1})$ for $s_t^{\star} = (s_t, W_t) \in \mathcal{S}^{\star}$ and $(s_t^{\star})_{t=0}^{T-1} \in (\mathcal{S}^{\star})^{T-1} \times \mathcal{X}$ as a valid trajectory in SG$^{\star}$. Under these conditions, recall that a forward policy abiding by the CB condition also satisfies the TB condition (da Silva et al., 2024a). Consequently, we henceforth assume that there is a $Z^{\star}$ such that

$$Z^{\star} \prod_{0 \leq t \leq T-1} p_F(s_{t+1}^{\star}|s_t^{\star}) = R(s_T) \prod_{0 \leq t \leq T-1} p_B(s_t^{\star}|s_{t+1}^{\star}) = R(s_T) \prod_{0 \leq t \leq T-1} \tilde{p}_B(s_t|s_{t+1}) \tag{16}$$

for each complete trajectory $(s_t^{\star})_{t=0}^{T}$ on the lifted DAG SG$^{\star}$. Also, recall that there are policies $p_S$ and $p_W$ such that

$$p_F(s_{t+1}^{\star}|s_t^{\star}) = p_S(s_{t+1}|s_t, W_t) \cdot p_W(W_{t+1}|s_{t+1}, W_t);$$

$p_W(W_{t+1}|s_{t+1}, W_t)$ is either 1 or 0—depending on whether $W_{t+1} = W_t + \phi(W_t, s_{t+1})$ or not. As Equation (16) holds only for trajectories in SG$^{\star}$, $p_W(W_{t+1}|s_t, W_t) = 1$ and

$$Z^{\star} \prod_{0 \leq t \leq T-1} p_S(s_{t+1}|s_t, W_t) = R(s_T) \prod_{0 \leq t \leq T-1} \tilde{p}_B(s_t|s_{t+1}). \tag{17}$$

We will show that $p_F(s_{t+1}^{\star}|s_t^{\star})$ is path-independent; equivalently, that $W_t$ does not depend on the path leading to $s_t^{\star}$. In doing so, we define $\tilde{p}_F(s_{t+1}|s_t) = p_F(s_{t+1}|s_t, W_t)$ for this particular $W_t$, concluding the demonstration. For this, let $\xi^{\star} = (s_t^{\star})_{t=0}^{l-1}$ be a (possibly incomplete) trajectory, $\xi = (\mathrm{s}_t)_{t=0}^{l-1}$, and $p_B(\xi^{\star}) := (R(s_T)/Z^{\star})p_B(\xi^{\star}|s_T)$ according to the factorization in Equation (16).

Importantly, the result is trivial for $l = 1$, as $W_o$ is fixed. From now on, assume $l \geq 2$. In this context, it is straightforward to see that

$$p_S(s_l|\xi^{\star}) = \frac{p_F(s_l^{\star}, \xi^{\star})}{p_F(\xi^{\star})} = \frac{\sum_{\tau \colon s_l^{\star} \rightsquigarrow \mathcal{X}} p_F(\xi^{\star}, s_l^{\star}, \tau)}{\sum_{\tau' \colon s_{l-1}^{\star} \rightsquigarrow \mathcal{X}} p_F(\xi^{\star}, \tau')};$$

$\tau \colon s_l^{\star} \rightsquigarrow \mathcal{X}$ refers to the space of trajectories starting at $s_l^{\star}$ and ending on any $x \in \mathcal{X}$. Consequently,

$$p_S(s_l|\xi^{\star}) = \frac{\sum_{\tau \colon s_l \rightsquigarrow \mathcal{X}} \tilde{p}_B(\xi, s_l, \tau)}{\sum_{\tau' \colon s_{l-1} \rightsquigarrow \mathcal{X}} \tilde{p}_B(\xi, \tau')}$$

by Equation (16); trajectories $\tau \colon s_l \rightsquigarrow \mathcal{X}$ inhabit the unlifted DAG. Since $\tilde{p}_B(\xi, s_l, \tau) = \tilde{p}_B(\xi|s_{l-1}) \cdot \tilde{p}_B(s_{l-1}, \tau)$ and $\tilde{p}_B(\xi, \tau') = \tilde{p}_B(\xi|s_{l-1}) \cdot \tilde{p}_B(s_{l-1}, \tau')$,

$$p_S(s_l|s_{l-1}, W_{l-1}) = p_S(s_l|\xi^{\star}) = \frac{\tilde{p}_B(\xi|s_{l-1}) \sum_{\tau \colon s_l \rightsquigarrow \mathcal{X}} \tilde{p}_B(s_{l-1}, \tau)}{\tilde{p}_B(\xi|s_{l-1}) \sum_{\tau' \colon s_{l-1} \rightsquigarrow \mathcal{X}} \tilde{p}_B(\tau')} = \tilde{p}_B(s_{l-1}|s_l) \frac{\sum_{\tau \colon s_l \rightsquigarrow \mathcal{X}} \tilde{p}_B(\tau)}{\sum_{\tau' \colon s_{l-1} \rightsquigarrow \mathcal{X}} \tilde{p}_B(\tau')} =: \tilde{p}_F(s_l|s_{l-1}).$$

That is, $p_S(s_l|\xi^{\star})$ depends only on $s_{l-1}$ and $s_l$—and not on the past latent dynamical system $\{W_t\}_{t \geq 0}$. This characterizes the Markovianity of $p_F$ (with respect to the natural filtration of the unlifted state space).

## B. Learning a recurrent flow

This section provides further details on the learning of a recurrent policy network—i.e., a policy $p_F$ on the lifted DAG of Definition D.1 satisfying either Corolalry 4.7 or 4.6. We also elaborate on the connection between our path-dependent algorithm and the flow network analogy that has been used to characterize GFlowNets.

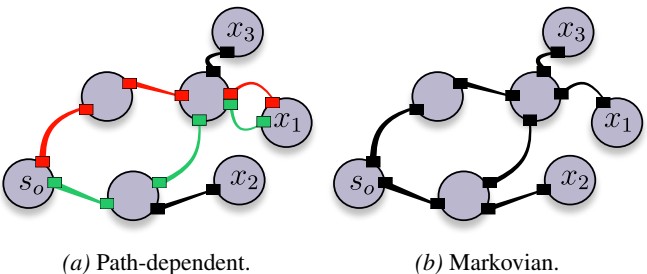

*(a) Path-dependent.*    *(b) Markovian.*

*Figure 8.* **Multi-gated flow network interpretation** for path-dependent (a) and Markovian (b) amortized samplers. Under our framework, each node sends a variable amount of flow to its children, depending on the flow's past trajectory (e.g., either green or red for $x_1$'s parent). In both cases, the flow reaching each terminal state $(x_1, x_2, x_3)$ is the same; see Section 4 for details.

**On the flow network analogy.** An important contribution of Bengio et al. (2021)'s seminal work on GFlowNets was the proposed physical interpretation of sampling as a flow assignment problem. Under this epistemology, the unnormalized distribution $R(x)$ of a terminal state $x \in \mathcal{X}$ is viewed as the amount of flow that should be transported from the initial state $(s_o)$ to $x$ through the edges of the state graph; see (Kim et al., 2024b, Figure 2) and Figure 8b. Interestingly, Figure 8a shows that our proposed framework can also be interpreted as learning a flow assignment in a *multi-gated* flow network, in which each node can send a variable amount of flow to its children depending on the value of the latent variable $W$ of the lifted state graph (Definition D.1). Drawing on this, the reader is invited to notice that many of the recent algorithmic improvements for GFlowNets (e.g., (Kim et al., 2024a; Malik et al., 2023; Lau et al., 2023; 2024; Kim et al., 2025a)) can be easily adapted to our setting. We discussed some of them in Section 4 in the main text and in Section E in the supplement (below).

**Learning a recurrent policy network.** For completeness, we also elaborate on the search for policies $p_F$ and $p_B$ satisfying any of the introduced balance conditions in . To achieve this, we simply minimize the expectation of the log-squared difference between the left- and right-hand sides in Proposition 4.5 via stochastic gradient descent. As the corresponding stochastic programming problems have already been introduced and studied elsewhere (Bengio et al., 2021; Malkin et al., 2022; 2023; Madan et al., 2022; da Silva et al., 2024a; Hu et al., 2024b; Zhang et al., 2023a) in the context of Markovian samplers, they are only briefly described in the following corollary of Proposition 4.5.

**Corollary B.1** (Learning objectives). *Let $p_E$ be an exploratory policy[2], which may depend on $p_F$ (e.g., an $\epsilon$-greedy policy: $p_E(\cdot|s) = (1-\epsilon)p_F(\cdot|s) + \epsilon p_U(\cdot|s)$, with $p_U$ as the uniform policy). Then, the unrestricted global optima of each of the following objective functions over all policies respectively satisfy the TB (Corollary 4.6), and CB (Corollary 4.7) conditions.*

1. *(TB loss). Let $Z = F(s_o^\star)$ be the initial state's flow. Then,*

$$\mathcal{L}_{\mathrm{TB}}(p_F, p_B, Z) = \mathbb{E}_{\tau \sim p_E}\left[\left(\log \frac{Z p_F(\tau|s_o^\star)}{p_B(\tau|x)R(x)}\right)^2\right]$$

2. *(CB loss). Due to the variance identity for i.i.d. squared differences (Richter et al., 2020; Zhang et al., 2023b),*

$$\mathcal{L}_{\mathrm{CB}}(p_F, p_B) = \mathbb{E}_{\tau, \tau' \sim p_E}\left[\left(\log \frac{p_F(\tau|s_o^\star)p_B(\tau'|x')R(x')}{p_B(\tau|x)p_F(\tau'|s_o^\star)R(x)}\right)^2\right]$$

$$= 2\mathrm{Var}_{\tau \sim p_E}\left[\log \frac{p_F(\tau|s_o^\star)}{p_B(\tau|x)R(x)}\right]$$

*in which $x, x'$ denote the terminal states of $\tau, \tau'$.*

Importantly, Corollary B.1 ensures that the transition dynamics for both $\{s_t\}_{t \geq 1}$ and $\{W_t\}_{t \geq 1}$ in Figure 1b can be jointly learned by minimizing a single stochastic objective. Also, it is worth noticing that the simplicity of the above formulation is only possible due to determinisitic nature of the latent transitions. When the $\{W_t\}_{t \geq 1}$ follows a stochastic rule, the state

---

[2]An *exploratory policy* is a policy assigning positive probability to each children of the state graph at every state. Under the formalism of Section D, this means that the corresponding kernel $P_E$ is mutually absolutely continuous with respect to base measure $\kappa_F$.

space ceases to be countable; as a consequence, a rigorous treatment of the resulting method depends on a more sophistcated mathematical formalism—see Section D. Intriguingly, our analysis intentionally avoids considering the sampling on non-acyclig environment, e.g., (Brunswic et al., 2024; Morozov et al., 2025). That said, our path-dependent framework can be naturally extended to this broader setting. In fact, we believe that the latent dynamical system can be naturally used for biasing the amortized sampler towards shorter trajectories in the context of cyclic generation—which is an interesting direction for future research.

## C. Related works

A systematic study of amortized neural samplers for discrete and compositional spaces was initiated by the introduction of Generative Flow Networks (GFlowNets; Bengio et al., 2021). Originally, GFlowNets were proposed as a diversity-seeking reinforcement learning algorithm (Bengio et al., 2021; 2023). Soon after Bengio et al. (2021)'s seminal work, GFlowNets were successfully applied to molecule and biological sequence discovery (Bengio et al., 2021; Jain et al., 2022; Wang et al., 2023; Nica et al., 2022), structure learning (Deleu et al., 2022; 2023), combinatorial optimization (Zhang et al., 2023a;b; 2025), phylogenetic inference (Zhou et al., 2024), and fine-tuning of large language models (Hu et al., 2024a; Venkatraman et al., 2024). In parallel, there has been a significant effort in developing algorithmic improvements for reducing GFlowNet's sample complexity. These approaches have focused on improved credit assignment via reward shaping (Pan et al., 2023b;a), more effective learning objectives (Madan et al., 2022; Deleu et al., 2024), and heuristics for enhancing exploration of high-probability regions (Kim et al., 2024b; 2025b) and shortening the trajectory length (Boussif et al., 2024; Silva et al., 2025b) for reduce complexity. Given Proposition 4.5, we notice that our method complements—rather than substitutes—these techniques. On top of this, there has also been a growing body of literature aimimg to build a principled framework for understanding GFlowNets through the lens of optimization (Yu, 2025), expressivity (Silva et al., 2025a), and generalization (Shen et al., 2023b; Atanackovic & Bengio, 2024). In (Silva et al., 2025a), in particular, it has been shown for the problems described Section 5 that the *flow-consistency in subgraphs* (FCS),

$$\text{FCS}(p, q) = \mathbb{E}_{\mathcal{B}:=\{x_1,\ldots,x_B\}\sim r}\left[\frac{1}{2}\sum_{1\leq i\leq B}\left|\frac{p(x_i)}{p(\mathcal{B})} - \frac{q(x_i)}{q(\mathcal{B})}\right|\right] = \mathbb{E}_{\mathcal{B}\sim r}\left[\text{TV}(p|_{\mathcal{B}}, q|_{\mathcal{B}})\right], \tag{18}$$

in which $p$, $q$, and $r$ are distributions, $B$ is a constant, and $p(\mathcal{B}) = \sum_{x\in\mathcal{B}} p(x)$, and $p|_{\mathcal{B}}$ is the restriction of $p$ on $\mathcal{B}$, provides a reliable measurement for the goodness-of-fit of GFlowNets, with a (Spearman) correlation of over $90\%$ with the TV distance between the learned and target distributions on the entire space; we followed the instructions in (Silva et al., 2025a, Section E) for evaluating FCS. Interestingly, our sampling algorithm is reminiscent of Gibbs sampling (Jones & Hobert, 2001; Gelfand & Smith, 1990), implementing a systematic scannig strategy for state modification: $(s, \omega) \to (s', \omega) \to (s', \omega')$.

## D. Stochastic latent dynamical system

This section builds on the framework of Lahlou et al. (2023) to expand our path-dependent algorithm in Figure 1b to the setting in which the latent dynamical system $\{W_t\}_{t\geq 1}$ evolves according to a stochastic rule. The resulting lifted DAG, defined below, is illustrated in Figure 9. Contrarily to our exposition in Section 4.1, in which we solely aimed to sample from a distribution over $\mathcal{X}$, we presently focus in sampling from a joint distribution $\pi(x, \omega)$ over $\mathcal{X}$ and a user-defined set $\Omega$ such that the marginal of $\pi$ over $\mathcal{X}$ matches our target distribution ($R$, in the main text).

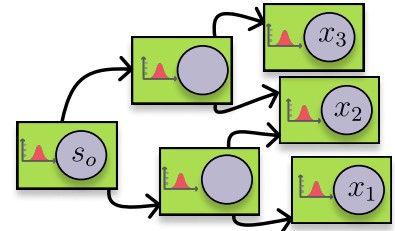

*Figure 9.* Illustration of Definition D.1. We endow each node in the unlifted DAG (recall Definition 2.1) with a latent and stochastic random variable (in red).

**Definition D.1.** Let $(\bar{\mathcal{S}}, \Sigma_{\mathcal{S}})$ and $(\Omega, \Sigma_\Omega)$ be measurable topological spaces with Borel $\sigma$-algebras $\Sigma$ and $\Sigma_\Omega$, respectively. Also, let $\nu$ be a positive probability measure on the product space $(\bar{\mathcal{S}} \times \Omega, \Sigma)$, in which $\Sigma$ is the $\sigma$-algebra induced by the product $\Sigma_{\mathcal{S}} \times \Sigma_\Omega$. Let $\mathcal{S}^\star = \bar{\mathcal{S}} \times \Omega$ be the *lifted state space*. We define $\bar{\mathcal{S}} = \{s_o\} \cup \mathcal{S} \cup \mathcal{X}$, in which $\{s_o\}$ and $\mathcal{X}$ are special open sets referred to respectively as *initial* and *terminal* states. We assume the existence of *forward* $\kappa_F \colon \mathcal{S}^\star \to \mathcal{M}(\Sigma)$ and *backward* $\kappa_B \colon \mathcal{S}^\star \to \mathcal{M}(\Sigma)$ kernels mapping each $s^\star \in \mathcal{S}^\star$ to a probability measure over $\Sigma$ and satisfying the following properties.

1. **(Duality).** $\nu(\mathrm{d}s)\kappa_F(s, \mathrm{d}s') = \nu(\mathrm{d}s')\kappa_B(s', \mathrm{d}s)$ on $\Sigma|_{(\{s_o\}\cup\mathcal{S})\times\Omega} \times \Sigma|_{(\mathcal{S}\cup\mathcal{X})\times\Omega}$.

2. **(Forward absorption).** There exists a $N \in \mathbb{N}$ for which $\kappa_F^N(s, \mathcal{X} \times \Omega) = 1$ for all $s \in (\{s_o\} \cup \mathcal{S}) \times \Omega$, in which

$\kappa_F^N(s, B) = \int_{\mathcal{S}^\star} \kappa_F(s, \mathrm{d}s') \kappa_F^{N-1}(s', B)$ is recursively defined. Also, $\kappa_F(x^\star, \{x\} \times \Omega) = 1$ $\nu$-almost surely for $x^\star \in \{x\} \times \Omega$ and $x \in \mathcal{X}$ and $\kappa_F(s, (\mathcal{S} \cup \mathcal{X}) \times \Omega) = 1$ for every $s \in \mathcal{S}^\star$.

3. **(Reachability).** For each measurable $B \in \Sigma$, there is an integer $k \in \mathbb{N}$ such that $\int_{\mathcal{S}_o^\star} \nu(\mathrm{d}s_o^\star) \kappa_F^k(s_o^\star, B) > 0$, in which $\mathcal{S}_o^\star = \{s_o\} \times \Omega$.

4. **(Backward absorption).** $\mathcal{S}_o^\star$ is the only absorbing set for $\kappa_B$, i.e., $\kappa_B(s_o^\star, \mathcal{S}_o^\star) = 1$ $\nu$-almost surely for $s_o^\star \in \mathcal{S}_o^\star$. Also, $\kappa_B(s^\star, (\bar{\mathcal{S}} \setminus \mathcal{X}) \times \Omega) = 1$ $\nu$-almost surely for $s^\star \in \mathcal{S}^\star$, in particular, $\kappa_B$ leaves $\mathcal{X}^\star$ $\nu$-almost surely.

5. **(Continuity).** Both $s \mapsto \kappa_F(s, B)$ and $s \mapsto \kappa_B(s, B)$ are continuous functions (wrt given topology) for all $B \in \Sigma$.

6. **(Absolute continuity).** Both $\kappa_F(s, \cdot)$ and $\kappa_B(s, \cdot)$ are absolutely continuous wrt $\nu$ for all $s \in \mathcal{S}^\star$.

7. **($\nu$-nondegeneracy).** For each $s \in \mathcal{S}^\star$, if $\kappa_B(s, E) > 0$ for some $E \in \Sigma$, then $\kappa_B(s, \tilde{E}) > 0$ for every measurable subset $\tilde{E} \subseteq E$ with $\nu(\tilde{E}) > 0$.

Definition D.1 characterizes a lifted DAG equipped with a continuous and stochastic latent variable inhabiting a set $\Omega$. Before proceeding, we recall that $\Sigma|_A$ is the restriction of the $\sigma$-algebra $\Sigma$ to the measurable set $A$; since $\sigma$-algebras are closed for intersection, this is well-defined. Given a measure $\xi$ over $\Sigma|_{\{s_o\} \times \Omega}$, our objective is that the marginal distribution over $\mathcal{X} \times \Omega$ induced by a (learned) kernel $P_F \colon \mathcal{S}^\star \to \mathcal{M}(\Sigma)$ absolutely continuous wrt $\kappa_F$ matches a given (perhaps unnormalized) target $\Pi \colon \Sigma|_{\mathcal{X} \times \Omega} \to [0, 1]$. In other words,

$$\int_{\{s_o\} \times \Omega} \xi(\mathrm{d}s_o^\star) P_F^N(s_o, B) = \Pi(B)$$

for each $B \in \Sigma|_{\mathcal{X} \times \Omega}$. When there exist probability measures $R \colon \Sigma_{\mathcal{S}}|_{\mathcal{X}} \to [0, 1]$ and $P_\Omega \colon \Sigma_\Omega \to [0, 1]$ such that $\Pi(B_{\mathcal{S}} \times B_\Omega) = R(B_{\mathcal{S}}) P_\Omega(B_\Omega)$ for $B_{\mathcal{S}} \times B_\Omega \in \Sigma$, the condition above ensures that the marginal distribution of $\xi(\mathrm{d}s_o^\star) \kappa_F(s_o^\star, B)$ over $\mathcal{X}$ when the latent variable in $\Omega$ is marginalized out matches $R(B)$. It remains to be answered how to learn such a $\kappa_F$ and $\xi$ and how to define $P_\Omega$. We will comment on these facts in passing below. Towards this objective, we first state a few general facts about the lifted continuous DAG in Definition D.1. We start by showing that $\mathcal{X} \times \Omega$ is a finite absorbing state for $\kappa_F$; in lieu of the statement below, we say that $\kappa_F$ is a finitely absorbing kernel with horizon $N$.

**Lemma D.2.** Let $\mathcal{X}^\star = \mathcal{X} \times \Omega$. Then, $\kappa_F^n(s, \mathcal{X}^\star) = 1$ for all $n \geq N$.

*Proof.* We proceed by simple induction. The base case, $\kappa_F^N(s, \mathcal{X}^\star) = 1$, follows from the definition. Assume that $\kappa_F^n(s, \mathcal{X}^\star) = 1$ for some $n \geq N$. Then,

$$\kappa_F^{n+1}(s, \mathcal{X}^\star) = \int_{\mathcal{S}^\star} \kappa_F(s, \mathrm{d}s') \kappa_F^n(s', \mathcal{X}^\star) = \int_{\mathcal{S}^\star} \kappa_F(s, \mathrm{d}s') = 1.$$

By induction, $\kappa_F^n(s, \mathcal{X}^\star) = 1$ for all $n \geq N$. $\qquad\qquad\square$

We also demonstrate that the duality in Definition D.1 holds for trajectories.

**Lemma D.3.** $\nu(\mathrm{d}s_1) \prod_{i=1}^n \kappa_F(s_i, \mathrm{d}s_{i+1}) = \nu(\mathrm{d}s_{n+1}) \prod_{i=1}^n \kappa_B(s_{i+1}, \mathrm{d}s_i)$ for each $n \geq 2$ on $\Sigma_{(\{s_o\} \cup \mathcal{S}) \times \Omega} \times \left( \bigtimes_{i=2}^n \Sigma_{\mathcal{S} \times \Omega} \right) \times \Sigma_{(\mathcal{S} \cup \mathcal{X}) \times \Omega}$.

*Proof.* We will show that this holds for $n = 2$, and then proceed inductively. For conciseness, let $A = \{s_o\} \cup \mathcal{S}$ and $B = \mathcal{S} \cup \mathcal{X}$. For any measurable function $f \colon K_2 \to \mathbb{R}$ for $K_2 = (A \times \Omega) \times (\mathcal{S} \times \Omega) \times (B \times \Omega)$,

$$\int_{K_2} f(s_1, s_2, s_3) \nu(\mathrm{d}s_1) \kappa_F(s_1, \mathrm{d}s_2) \kappa_F(s_2, \mathrm{d}s_3) = \int_{K_2} f(s_1, s_2, s_3) \nu(\mathrm{d}s_2) \kappa_B(s_2, \mathrm{d}s_1) \kappa_F(s_2, \mathrm{d}s_3)$$

$$= \int_{K_2} f(s_1, s_2, s_3) \nu(\mathrm{d}s_3) \kappa_B(s_3, \mathrm{d}s_2) \kappa_B(s_2, \mathrm{d}s_1)$$

by duality. Inductively, assume that the identity holds for $n$. Then, let $f\colon K_{n+1} \to \mathbb{R}$ be any real-valued measurable function, in which $K_{n+1} = (A \times \Omega) \times (\mathcal{S} \times \Omega)^n \times (B \times \Omega)$. As such,

$$\int_{K_{n+1}} f(s_1, \ldots, s_{n+2}) \nu(\mathrm{d}s_1) \prod_{i=1}^{n+1} \kappa_F(s_i, \mathrm{d}s_{i+1}) = \int_{K_{n+1}} f(s_1, \ldots, s_{n+2}) \nu(\mathrm{d}s_{n+1}) \prod_{i=1}^{n} \kappa_B(s_{i+1}, \mathrm{d}s_i) \kappa_F(s_{n+1}, \mathrm{d}s_{n+2})$$

$$= \int_{K_{n+1}} f(s_1, \ldots, s_{n+2}) \nu(\mathrm{d}s_{n+2}) \prod_{i=1}^{n+1} \kappa_B(s_{i+1}, \mathrm{d}s_i),$$

in which the first equality follows from the induction hypothesis (and from the measurability of $f_{s_{n+2}}\colon K_n \to \mathbb{R}$, $f_{s+2}(s_1, \ldots, s_{n+1}) := f(s_1, \ldots, s_{n+2})$) and the second, from the duality between $\kappa_F$ and $\kappa_B$ in $\Sigma|_{\mathcal{S} \times \Omega} \times \Sigma|_{(\mathcal{S} \cup \mathcal{X}) \times \Omega}$. As $f$ was chosen arbitrarily, we verified that $\nu(\mathrm{d}s_1) \prod_{i=1}^{n} \kappa_F(s_i, \mathrm{d}s_{i+1}) = \nu(\mathrm{d}s_{n+1}) \prod_{i=1}^{n} \kappa_B(s_{i+1}, \mathrm{d}s_i)$ for each $n \geq 2$ on $\Sigma_{(\{s_o\} \cup \mathcal{S}) \times \Omega} \times (\bigtimes_{i=2}^{n} \Sigma_{\mathcal{S} \times \Omega}) \times \Sigma_{(\mathcal{S} \cup \mathcal{X}) \times \Omega}$. $\qquad\square$

In view of Lemma D.3, we recursively define the operation $\kappa_F^{\circ n}(s, A)$ for $s \in (\{s_o\} \cup \mathcal{S}) \times \Omega$ and $A \in \Sigma|_{(\mathcal{S} \cup \mathcal{X}) \times \Omega}$ as

$$\kappa_F^{\circ n}(s, A) = \int_{(\mathcal{S} \times \Omega)} \kappa_F^{\circ(n-1)}(s, \mathrm{d}s') \kappa_F(s', A),$$

and analogously for $\kappa_B$. Under these conditions, $\nu(\mathrm{d}s) \kappa_F^{\circ n}(s, \mathrm{d}s') = \nu(\mathrm{d}s') \kappa_B^{\circ n}(s', \mathrm{d}s)$ on $\Sigma_{(\{s_o\} \cup \mathcal{S}) \times \Omega} \times \Sigma_{(\mathcal{S} \cup \mathcal{X}) \times \Omega}$. Clearly, $\kappa_F^n(s, A) \geq \kappa^{\circ n}(s, A)$ for each $n$, $s$, and $A$, and the reachability condition in Definition D.1 implies that there is a $k$ for each measurable $E \subseteq \mathcal{S}^\star$ such that $\int_{\mathcal{S}_o^\star} \nu(\mathrm{d}s_o^\star) \kappa_F^{\circ k}(s_o^\star, E) > 0$. We will show that $\kappa_B^{\circ n}(s, \cdot)$ is both absolutely continuous and non-degenerate with respect to $\nu$.

**Lemma D.4.** *For each integer $n \geq 1$ and $s \in \mathcal{S}^\star$, $\kappa_B^{\circ n}(s, \cdot)$ is $\nu$-nondegenerate and absolutely continuous with respect to $\nu$.*

*Proof.* We proceed by induction. Define $\kappa_B^{\circ 1}(s, \cdot) := \kappa_B(s, \cdot)$. By definition, $\kappa_B(s, \cdot)$ is both absolutely continuous and non-degenerate with respect to $\nu$. Assume that $\kappa^{\circ n}(s, \cdot)$ also satisfies these properties for a $n \geq 1$. Then, for each measurable $A$,

$$\kappa_B^{\circ n+1}(s, A) := \int_{\mathcal{S} \times \Omega} \kappa_B(s, \mathrm{d}s') \kappa_B^{\circ n}(s', A).$$

For absolute continuity, notice that $\nu(A) = 0$ implies $\kappa_B^{\circ n}(s, A) = 0$ by our induction hypothesis. Hence, $\kappa_B^{\circ n+1}(s, A) = 0$. That is, $\kappa_B^{\circ n+1}(s, \cdot)$ is absolutely continuous with respect to $\nu$. For nondegeneracy, let $E \subseteq \mathcal{S}^\star$ measurable be such that $\kappa_B^{\circ n+1}(s, E) > 0$. Then, there is a set $D$ with positive $\kappa_B(s, \cdot)$-measure such that $\kappa_B^{\circ n}(s', E) > 0$ for every $s' \in D$. By induction hypothesis, if $\tilde{E} \subseteq E$ is a measurable subset of $E$ such that $\nu(\tilde{E}) > 0$, then $\kappa_B^{\circ n}(s', \tilde{E}) > 0$ for each $s' \in D$. Consequently, $\kappa_B^{\circ n+1}(s, \tilde{E}) > 0$. Inductively, we have shown that $\kappa_B^{\circ n+1}(s, \cdot)$ is $\nu$-nondegenerate. $\qquad\square$

The next lemma shows that, when starting at $\mathcal{X} \times \Omega$, a Markov chain following $\kappa_B$ reaches $\{s_o\} \times \Omega$ in at most $N$ steps. The intuition is that, were this not the case, the chain would arrive at some set $B \in \Sigma|_{\mathcal{S} \times \Omega}$ in $N$ steps with positive probability. As each set $B$ is reachable from $\{s_o\} \times \Omega$ with positive probability, we would by duality be able to construct a $\kappa_F$-based Markov chain starting at $\{s_o\} \times \Omega$ that would, with positive probability, remain outside $\mathcal{X} \times \Omega$ after $N$ steps. This contradicts the above lemma. As a consequence, $\kappa_B$ is also a finitely absorbing kernel with horizon $N$.

**Lemma D.5.** $\kappa_B^N(x^\star, \{s_o\} \times \Omega) = 1$ *for each $x^\star \in \{x\} \times \Omega$ and $x \in \mathcal{X}$.*

*Proof.* Assume for contradiction that $\kappa_B^N(x^\star, \mathcal{S}_o^\star) < 1$ for some $x^\star$. We will show that there is a positive-measure at-least-$N$-sized trajectory from $\mathcal{S}_o^\star$ to a subset of $\mathcal{S}^\star \setminus \mathcal{X}^\star$. By Lemma D.2, however, such a trajectory cannot exist. To start with, Lemma D.3 ensures that $\kappa_B^{\circ N}(x^\star, \mathcal{S}_o^\star) \leq \kappa_B^N(x^\star, \mathcal{S}_o^\star) < 1$. As $\kappa_B^{\circ N}(x^\star, \mathcal{S}_o^\star) < 1$, $\kappa_B^{\circ N}(x^\star, (\mathcal{S}^\star \setminus \mathcal{S}_o^\star)) > 0$. By definition,

$$\kappa_B^{\circ N}(x^\star, (\mathcal{S}^\star \setminus \mathcal{S}_o^\star)) := \int_{\mathcal{S} \times \Omega} \kappa_B(x^\star, \mathrm{d}s) \kappa_B^{\circ N-1}(s, \mathcal{S}^\star \setminus \mathcal{S}_o^\star) = \int_{\mathcal{S} \times \Omega} \kappa_B(x^\star, \mathrm{d}s) \kappa_B^{\circ N-1}(s, \mathcal{S}^\star \setminus \mathcal{S}_o^\star) > 0.$$

As $s \mapsto \kappa_B^{\circ N-1}(s, \mathcal{S}^\star \setminus \mathcal{S}_o^\star))$ is a non-negative and continuous function, the equation above implies that there is an open set with positive $\kappa_B(x^\star, \cdot)$-measure $D$ for which $\kappa_B^{\circ N-1}(s, \mathcal{S}^\star \setminus \mathcal{S}_o^\star) > 0$ for all $s \in D$. By the (contraposition of) absolute continuity, $D$ has also $\nu$-positive measure. Also, $D \subseteq \mathcal{S} \times \Omega$ due to the integration domain. Fix $s \in D$. Then, there is an measurable set $E \subseteq \mathcal{S}$ for which $\kappa_B^{\circ N-1}(s, E) > 0$. By continuity, there is a neighborhood $\tilde{D} \subseteq D$ of $s$ for which $\kappa_B^{\circ N-1}(s, E) > 0$

for all $s \in \tilde{D}$. By reachability, there is a $k$ for which $\int_{\mathcal{S}_o^\star} \nu(\mathrm{d}s_o^\star) \kappa_F^{\circ k}(s_o^\star, E) > 0$. By Lemma D.3, since $E \subseteq (\mathcal{S} \cup \mathcal{X}) \times \Omega$,

$$\int_{\mathcal{S}_o^\star} \nu(\mathrm{d}s_o^\star) \kappa_F^{\circ k}(s_o^\star, E) = \int_E \nu(\mathrm{d}s) \kappa_B^{\circ k}(s, \mathcal{S}_o^\star) > 0.$$

Again, by the non-negativity and continuity of $s \mapsto \kappa_B^{\circ k}(s, \mathcal{S}_o^\star)$, there is a set $\tilde{E} \subseteq E$ for which $\kappa_B^{\circ k}(s, \mathcal{S}_o^\star) > 0$ for all $s \in \tilde{E}$, and $\tilde{E}$ has positive $\nu$-measure. In other words, a Markov chain following $\kappa_B$ moves from $\tilde{D}$ to $\tilde{E}$ in $N-1$ steps, and from $\tilde{E}$ to $\mathcal{S}_o^\star$ in $k \geq 1$ steps (recall that $\kappa_B^{\circ N-1}(s, \tilde{E}) > 0$ for all $s \in \tilde{D}$ since $\tilde{E}$ is a measurable subset of $E$ with positive $\nu$-measure—by $\nu$-nondegeneracy; see Lemma D.4). By duality and Lemma D.3,

$$
\begin{aligned}
0 = \int_{\mathcal{S}_o^\star} \nu(\mathrm{d}s_o^\star) \kappa_F^{N+k-1}(s_o^\star, \tilde{D}) &\geq \int_{\mathcal{S}_o^\star} \nu(\mathrm{d}s_o^\star) \kappa_F^{\circ N+k-1}(s_o^\star, \tilde{D}) \\
&= \int_{\tilde{D}} \nu(\mathrm{d}s^\star) \kappa_B^{\circ N+k-1}(s^\star, \mathcal{S}_o^\star) \\
&= \int_{\tilde{D}} \nu(\mathrm{d}s^\star) \int_{\mathcal{S}} \kappa_B^{\circ N-1}(s^\star, \mathrm{d}s^{\star\prime}) \kappa_B^{\circ k}(s^{\star\prime}, \mathcal{S}_o^\star) \\
&\geq \int_{\tilde{D}} \nu(\mathrm{d}s^\star) \int_{\tilde{E}} \kappa_B^{\circ N-1}(s^\star, \mathrm{d}s^{\star\prime}) \kappa_B^{\circ k}(s^{\star\prime}, \mathcal{S}_o^\star) > 0,
\end{aligned}
$$

since $\kappa_B^{\circ k}(s, \mathcal{S}_o^\star)$ is a strictly positive function on $\tilde{E}$ and both $\tilde{D}$ and $\tilde{E}$ are positive sets wrt their corresponding measures. The initial equality follows from Lemma D.2 since $\tilde{D} \in \mathcal{S}^\star \setminus \mathcal{X}^\star$. This contradiction ensures that $\kappa_B^N(x^\star, \mathcal{S}_o^\star) = 1$ for all $x^\star$. $\square$

Similarly to Lemma D.2, $\kappa_B^{N+k}(x^\star, \mathcal{S}_o^\star) = 1$ for each $k \geq 0$. This is the content of the next lemma.

**Lemma D.6.** *Let $\mathcal{S}_o^\star = \{s_o\} \times \Omega$. Then, $\kappa_B^n(x^\star, \mathcal{S}_o^\star) = 1$ for all $x^\star \in \mathcal{X}^\star$ and $n \geq N$.*

*Proof.* We proceed as in Lemma D.2—by induction. By Lemma D.5, $\kappa_B^N(x^\star, \mathcal{S}_o^\star) = 1$. Assume that $\kappa_B^n(x^\star, \mathcal{S}_o^\star) = 1$ for some $n \geq N$. Then,

$$\kappa_B^{n+1}(x^\star, \mathcal{S}_o^\star) = \int_{\mathcal{S}^\star} \kappa_B^n(x^\star, \mathrm{d}s) \kappa_B(s, \mathcal{S}_o^\star) = \int_{\mathcal{S}_o^\star} \kappa_B^n(x^\star, \mathrm{d}s) \kappa_B(s, \mathcal{S}_o^\star).$$

As $\mathcal{S}_o^\star$ is an absorbing set for $\kappa_B$, $\kappa_B(s, \mathcal{S}_o^\star) = 1$ for each $s \in \mathcal{S}_o^\star$ $\nu$-almost surely. Due to absolute continuity, $\kappa_B(s, \mathcal{S}_o^\star) = 1$ $\kappa_B(x^\star, \cdot)$-almost surely for $s \in \mathcal{S}_o^\star$. Consequently,

$$\kappa_B^{n+1}(x^\star, \mathcal{S}_o^\star) = \int_{\mathcal{S}_o^\star} \kappa_B^n(x^\star, \mathrm{d}s) \kappa_B(s, \mathcal{S}_o^\star) = \int_{\mathcal{S}_o^\star} \kappa_B(x^\star, \mathrm{d}s) = \kappa_B(x^\star, \mathcal{S}_o^\star) = 1.$$

Inductively, $\kappa_B^n(x^\star, \mathcal{S}_o^\star) = 1$ for all $n \geq N$. $\square$

Notably, Definition D.1 and the ensuing lemmas provide a complete characterization of a lifted pointed DAG with a continuous latent state. Curiously, our assumptions in Definition D.1 differ slightly from that of Lahlou et al. (2023, Definition 1), for instance, we assume that $\{s_o\}$ is an absorbing set for $\kappa_B$, while Lahlou et al. (2023) establishes that $\kappa_B(s_o, \cdot)$ is the trivial (null) measure. Below, we explain how an amortized sampler over the lifted space in Definition D.1 can be learned. Aftewards, we elaborate on the instantiation of our proposed framework for the LINES environment described in Section 3.

**Learning an amortized sampler.** In pratice, we learn kernels $P_F(s, \cdot)$ and $P_B(s, \cdot)$ absolutely continuous with respect to $\kappa_F(s, \cdot)$ and $\kappa_B(s, \cdot)$, respectively, and a measure $\xi$ on $\mathcal{S}_o^\star$ absolutely continuous with respect to $\nu$. Given a target measure $\Pi$ over $\mathcal{X}^\star$ absolutely continuous with respect to $\nu$, our objective is to search for a triple of $\xi$, $P_F$, and $P_B$ satisfying

$$\xi(\mathrm{d}s_o^\star) P_F^N(s_o^\star, \mathrm{d}x) = \Pi(\mathrm{d}x) P_B^N(x, \mathrm{d}s_o^\star).$$

In doing so, we verify that, for each measurable $B$,

$$\int_{S_o^\star} \xi(\mathrm{d}s_o^\star) P_F^N(s_o^\star, B) = \int_B \Pi(\mathrm{d}x) P_B^N(x, \mathcal{S}_o^\star).$$

Since $P_B(x, \cdot) \ll \kappa_B(x, \cdot)$, an argument similar to that of Lemma D.4 shows that $P_B^N(x, \cdot) \ll \kappa_B^N(x, \cdot)$. By Lemma D.5, $\kappa_B^N(x, \mathcal{S}^\star) = 1$; hence, $\kappa_B^N(x, (\mathcal{S}_o^\star)^c) = 0$ and $P_B^N(x, (\mathcal{S}_o)^c) = 0$. That is, $P_B^N(x, (\mathcal{S}_o^\star)^c) = 1$ since $P_B^N(x, \cdot)$ is a probability measure. Thus,

$$\int_{\mathcal{S}_o^\star} \xi(\mathrm{d}s_o^\star) P_F^N(s_o^\star, B) = \int_B \Pi(\mathrm{d}x) P_B^N(x, \mathcal{S}_o^\star) = \Pi(B).$$

From an implementation perspective, we represent $P_F$, $P_B$, $\xi$, and $\Pi$ with their densities $p_F$, $p_B$, $\pi$, and $\xi$ relatively to $\kappa_F$, $\kappa_B$, $\nu$, and $\nu$, respectively. When $\pi$ is known only up to a normalizing constant, $\pi(x) = Z\tilde{\pi}(x)$, we also estimate $Z$. By parameterizing $p_F$, $P_B$, $\xi$, and $Z$ with neural networks, learning is based on the minimization of the expected trajectory balance loss,

$$\mathcal{L}(s_o, s_1, \ldots, s_N) = \left( \log \xi(s_o) + \log \prod_{0 \le i \le N-1} p_F(s_i, s_{i+1}) - \log Z - \log \tilde{\pi}(s_N) - \log \prod_{0 \le i \le N-1} p_B(s_{i+1}, s_i) \right)^2,$$

with respect to a *exploratory policy* $P_E$ (absolutely continuous with respect to $\kappa_F$) via stochastic gradient descent. By Lemma D.2, $s_N \in \mathcal{X}^\star$ almost surely due to $P_E \ll \kappa_F$.

In conclusion, we also present a simple instantiation of Definition D.1 for concreteness.

*Example* D.7 (Stochastic latent dynamical system for the LINES environment). Illustratively, consider $\Omega = \mathbb{R}^d$, $\mathcal{S} = \mathcal{P} \setminus \{p_o\}$, $\mathcal{X} = \mathcal{Q}$, and $s_o = p_o$ in Figure 2 with step size $M = 1$. Under these conditions, $\mathcal{S}$ is a topological measurable space equipped with the discrete topology, while $\Omega$ is a Euclidean space endowed with the Lebesgue measure. Also, $\Sigma$ is the $\sigma$-algebra induced by the product of the $\sigma$-algebras of these measurable spaces, which is uniquely defined since both $\mathcal{S}$ and $\Omega$ are $\sigma$-finite. Naturally, for $s^\star = (p_i, \omega) \in \mathcal{S}^\star$, $\kappa_F(s^\star, \cdot)$ is supported on $\{q_i, p_{i+1}\} \times \mathbb{R}^d$, and $\kappa_B(s^\star, \cdot)$ is supported on $\{p_i\} \times \mathbb{R}^d$. In pratice, we could parameterize $p_F$ at $s_i^\star = (s_i, \omega_i)$ and $s_{i+1}^\star = (s_{i+1}, \omega_{i+1})$ as

$$\log p_F(s_i^\star, s_{i+1}^\star) = \log p_{F,\omega}((s_{i+1}, \omega_i), \omega_{i+1}) + \log p_{F,s}((s_i, \omega_i), s_{i+1}),$$

in which $p_{F,\omega}((s_{i+1}, \omega_i), \cdot) = \mathcal{N}(\phi(s_{i+1}, \omega_i), \sigma^2)$ is parameterized as a Gaussian distribution and $p_{F,s}((s_i, \omega_i), s_{i+1}) = \text{Categorical}(\psi(s_i, \omega_i))$, with $\phi$ and $\psi$ as neural networks. Similarly, $\xi(s_o)$ could be parameterized as a (mixture of) Gaussian(s), similarly to Lahlou et al. (2023). On top of that, $\pi(x, \omega) = Zr(x) \cdot \mathcal{N}(\omega | \mu, \Lambda)$, with $r$ as an unnormalized target distribution. Intuitively, this Gaussian-extended distribution would smooth up the landscape of the target and potentially facilitate the learning of an accurate sampler. That said, we leave a comprehensive investigation of this effect to future endeavors.

## E. Distributed and Streaming Inference in Path-dependent Models

The composable nature of amortized discrete samplers (Garipov et al., 2023) allows for the implementation of effective divide-and-conquer algorithms for streaming and distributed learning (da Silva et al., 2024a). In this section, we expand on the analysis in Section B to show that the existing techniques for non-centralized learning of amortized Markovian samplers can be seamlessly adapted to our path-dependent setting. Towards this objective, we recall the core assumption of such methods.

**Assumption E.1** (Factorizable target distribution). Let $\{R_t\}_{t \ge 1}$ be a sequence of functions $R_t \colon \mathcal{X} \to \mathbb{R}_+$. We assume that the target distribution, $R \colon \mathcal{X} \to \mathbb{R}_+$, can always be represented as $R(x) := \prod_{t=1}^T R_t(x)$ for some $T \in \mathbb{N}$.

Assumption E.1 simply states that the target distribution can be factorized into the product of independent functions. In the context of Bayesian inference, $\mathcal{X}$ might be thought of as the parameter space and $\{R_t\}_{t \ge 1}$ as the sequence of *sub-posteriors* induced by a independent stream of data $\{\mathcal{D}_t\}_{t \ge 1}$, i.e., $R_1(x) := \ell(\mathcal{D}_1 | x)p(x)$ and $R_t(x) := \ell(\mathcal{D}_t | x)$ for $t > 1$, in which $\ell$ is a likelihood function and $p$ is a prior distribution. When each $R_t$ is expensive to evaluate (e.g., due to the size of $\mathcal{D}_t$), learning a sampler for the full posterior $R(x) := \prod_{t=1}^T R_t(x)$ can be infeasible. Under these conditions, prior work has demonstrated that parallely learning small samplers for each $R_t$ and subsequently combining them in a centralized fashion can be done efficiently. Nextly, we demonstrate that this approach can be naturally accommodated into our path-dependent framework.

**Proposition E.2** (Distributed Amortized Sampling). *Let $\{R_t\}_{1 \le t \le T}$ be a sequence of $T$ target distributions. Also, let $(p_F^{(t)}, p_B^{(t)}, R_t)_{t=1}^T$ be a set of path-dependent amortized samplers abiding by Proposition 4.5 (with respect to their respective targets). Assume that $(p_F, p_B)$ satisfies*

$$\left( \frac{p_F(\tau | s_o^\star)}{p_B(\tau | x)} \right) \left( \prod_{1 \le t \le T} \frac{p_F^{(t)}(\tau' | s_o^\star)}{p_B^{(t)}(\tau' | x')} \right) = \left( \prod_{1 \le t \le T} \frac{p_F^{(t)}(\tau | s_o^\star)}{p_B^{(t)}(\tau | x)} \right) \left( \frac{p_F(\tau' | s_o^\star)}{p_B(\tau' | x)} \right) \tag{19}$$

*for each pair of complete trajectories $(\tau, \tau')$ ending at $(x, x')$ on the lifted DAG (see Definition D.1). Then, the marginal distribution of $p_F$ over $\mathcal{X}$ matches $\prod_{t=1}^{T} R_t$ up to a constant, i.e., $\sum_{\tau \rightsquigarrow x} p_F(\tau|s_o^\star) \propto \prod_{t=1}^{T} R_t(x)$ for each $x \in \mathcal{X}$.*

**Proposition E.3** (Streaming Amortized Sampling). *Let $\{R_t\}_{t \geq 1}$ be an unbounded sequence of target distributions. On top of that, let $\{(p_F^{(t)}, p_B^{(t)}, R_t)\}_{t \geq 1}$ be a sequence of amortized samplers recursively abiding by*

$$\left( \frac{p_F^{(T)}(\tau|s_o^\star)}{p_B^{(T)}(\tau|x)} \right) \left( R_T(x') \cdot \frac{p_F^{(T-1)}(\tau'|s_o^\star)}{p_B^{(T-1)}(\tau'|x')} \right) = \left( R_T(x) \cdot \frac{p_F^{(T-1)}(\tau|s_o^\star)}{p_B^{(T-1)}(\tau|x)} \right) \left( \frac{p_F^{(T)}(\tau'|s_o^\star)}{p_B^{(T)}(\tau'|x)} \right) \tag{20}$$

*for each $T > 1$ and trajectory pairs $(\tau, \tau')$ ending at $(x, x')$, and assume that $(p_F^{(1)}, p_B^{(1)}, R_1)$ satisfies the condition in Proposition 4.5. Then, the marginal distribution of each $p_F^{(T)}$ over $\mathcal{X}$ matches $\prod_{t=1}^{T} R_t$ for $T \geq 1$ (see Proposition E.3).*

Both Propositions E.2 and E.3 follow directly from Proposition 4.5. In particular, Proposition E.3 can be verified through a simple induction argument. From a learning perspective, Proposition E.2 suggests an divide-and-conquer algorithm for sampling from a target distribution $R$ satisfying Assumption E.1: we first train each $p_F^{(t)}$ parallely for $1 \leq t \leq T$, and subsequently learn a $p_F$ by minimizing the expected log-squared difference between the left- and right-hand-sides in Equation (19). Similarly, Proposition E.3 highlights that an iterative algorithm that updates the current model $(p_F^{(T-1)}, p_B^{(T-1)})$ according to Equation (20) results in an unbiased sampler for the product target distribution. In doing so, we avoid repeated evaluations of $R_t$ for each $t < T$ when learning an amortized sampler in a streaming fashion.

# F. Additional Experiments & Further Details

This section provides further details on the implementation for our experiments, along with a set of additional experiments for confirming our hypothesis that the path-dependent parameterization results in a better goodness-of-fit to the target distribution.

## F.1. Implementation details

**Computer code.** We provide computer code for reproducing our experiments in the supplement. We wrote both the environment and the deep neural networks based on highly-optimized JAX primitives (Bradbury et al., 2018). Our experiments were executed in a computer equipped with a 64GB AMD MI210 GPU.

**Models, optimizer, hyperparameters.** We used standard hyperparameters for our experiments (Viviano et al., 2025). In particular, in Section 5, our Markovian models were parameterized with 3-layer (leaky) ReLU networks. The hidden layer's size of both the Markovian and the path-dependent was chosen to ensure that the resulting neural networks would be similarly sized (in terms of the number of parameters). We used AdamW (Loshchilov & Hutter, 2019) with a learning rate of $10^{-3}$ for the policy's parameters and, when needed, a learning rate of $10^{-1}$ for the $\log Z$ parameter. Also, we minimized the variance loss for the sequence- and set-based environments, and the TB loss otherwise (see Corollary B.1), for estimating the parameters of the policies. We followed the experimental setup of Silva et al. (2025a) for the graph-structured tasks in Section 3.

**Defining the SRWM.** We used the normalized ELU link function $\sigma$ described in Equation (23) and, to mitigate per-step computation cost, used a smaller ($d = 32$) dimension for the latent dynamical system, the output of which went through a single-layer perceptron. The reason for this is that the the cost of our SRWM is dominated by the $\mathcal{O}(d^3)$ matrix multiplication required by the rotation matrix. More specifically, let $\mathbf{W}_t$ and $\mathbf{y}_t = \mathbf{W}_t \psi(s_t)$ be as in Equation (9). To compute $p_F(\cdot|s_t, \mathbf{W}_t)$, we introduce $\mathbf{W}_{\text{MLP},1} \in \mathbb{R}^{d_{\text{MLP}} \times d}$ and $\mathbf{W}_{\text{MLP},2} \in \mathbb{R}^{d_{\text{MLP}} \times d_{\text{out}}}$ and evaluate $p_F(\cdot|s_t, \mathbf{W}_t)$ as

$$\text{Softmax}(\mathbf{W}_{\text{MLP},2} \text{LeakyReLU}(\mathbf{W}_{\text{MLP},1} \mathbf{y}_t) \odot \mathbf{m}(s_t)),$$

in which we set $\alpha = 10^{-2}$ for LeakyReLU$(x)$, $d_{\text{out}}$ is given by the environment, and $\mathbf{m}(s_t) \in \{1, -\infty\}^{d_{\text{out}}}$ is a masking function delineating the permissible transitions at $s_t$—as in Equation (3). Importantly, we empirically observed that reducing $d$ negligibly affected learning convergence, but substantially improved computation efficiency.

## F.2. Additional experiments

We complement our analysis in Section 5 with further experiments on standard tasks for GFlowNets.

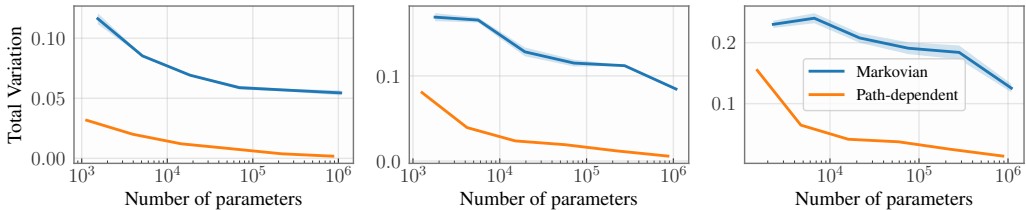

*Figure 10.* **TV distance to the target for varying model sizes** for the sequence environment. We show that our path-dependent parameterization based on the modified SRWM architecture consistently outperforms its Markovian counterpart (based on a 3-layer ReLU network). In each case, we increased the parameter count by expanding the latent dimension—e.g., $d$ in Equation (9). See Section 5.

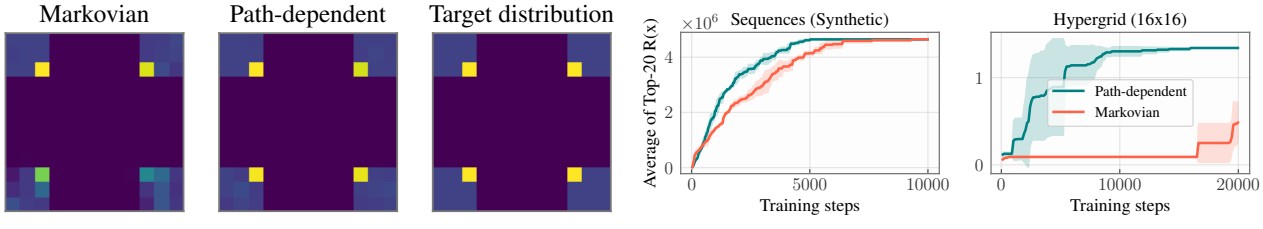

*(a)* Results for the standard hypergrid.   *(b)* Mode discovery for each parameterization.

*Figure 11.* (left) Our path-dependent parameterization improves goodness-of-fit for the standard $16 \times 16$ hypergrid task. (right) By learning a latent dynamical system, we also improve exploration of high-probability regions, as measured by the average $R(x)$ of the 20 most probable encountered states during training. (We consider the Syn-8 environment introduced in Table 2 in (b)).

**Preference learning (da Silva et al., 2024b).** We consider the problem of approximating a posterior distribution over the parameters of an integer-valued linear model for preference learning (Cole, 1993). Briefly, each object $i$ has a set of binary features, $\mathbf{x}_i \in \{1, 0\}^d$, and the probability of object $i$ being preferred over $j$ is modelled by $p(i \succ j | \mathbf{w}) = \sigma \left( \mathbf{w}^T (\mathbf{x}_i - \mathbf{x}_j) \right)$, with $\mathbf{w} \in \{-m, -m+1, \ldots, m\}$ for some positive integer $m$. We set an uniform prior over $\mathbf{w}$. As we show in Table 3 in the supplement, our model also provides a better approximation than its Markovian counterpart for this problem for different sample sizes (we fix both $d = 8$ and $m = 4$).

*Table 3.* Goodness-of-fit for a Markovian and path-dependent parameterizations for the preference learning task in terms of the TV distance to the target.

| Sample size | Markovian | Path-dependent |
|---|---|---|
| 50 | $0.135 \pm 0.022$ | $\mathbf{0.097} \pm 0.016$ |
| 100 | $0.177 \pm 0.035$ | $\mathbf{0.128} \pm 0.046$ |
| 150 | $0.274 \pm 0.074$ | $\mathbf{0.236} \pm 0.092$ |

**GridWorld (Bengio et al., 2021).** We show in Figure 11a a 2-dimensional grid environment with a target distribution defined for a point $(x, y)$ by

$$R(x, y) = 10^{-3} + 2 \cdot r_1(x) \cdot r_1(y) + 0.5 \cdot r_2(x) \cdot r_2(y),$$

in which $r_1(x) := [n(x) > 0.6] \cdot [n(x) > 0.8]$, $r_2(x) = [x > 0.5]$, and $n(x) = |2x/H-1 - 1|$ for a $H$-sized grid; $\cdot \mapsto [\cdot]$ is Inverson bracket, which corresponds to 1 if the evaluated expression is true and 0 otherwise. This $R$ is a standard in GFlowNet evaluation, e.g., (Malkin et al., 2023; Pan et al., 2023b). Notably, our path-dependent model also provides a better goodness-of-fit to the target in this scenario—which corroborates our observations in Section 5. On top of this, Figure 11 also indicates our approach improves state space exploration.

**Bit sequences.** We have also run experiments on the generation of bit sequences, which is a standard benchmark for GFlowNets (e.g., Malkin et al. (2022); Tiapkin et al. (2024); Silva et al. (2025b)). Towards this objective, we considered sequences of size $S \in \{16, 32, 64\}$. We adopted the target distribution proposed in Malkin et al. (2022, Section 5.3.1), and followed the experimental setup in (Silva et al., 2025b). Results in Table 4 show that our approach also provides a better goodnesss-of-fit to the target distribution for this task.

*Table 4.* Distance to the target distribution for the task of bit sequence generation.

| Seq. Size | Markovian | Path-dependent |
|---|---|---|
| 16 | $0.025 \pm 0.001$ | $\mathbf{0.001} \pm 0.000$ |
| 32 | $0.057 \pm 0.001$ | $\mathbf{0.002} \pm 0.000$ |
| 64 | $0.064 \pm 0.005$ | $\mathbf{0.002} \pm 0.000$ |

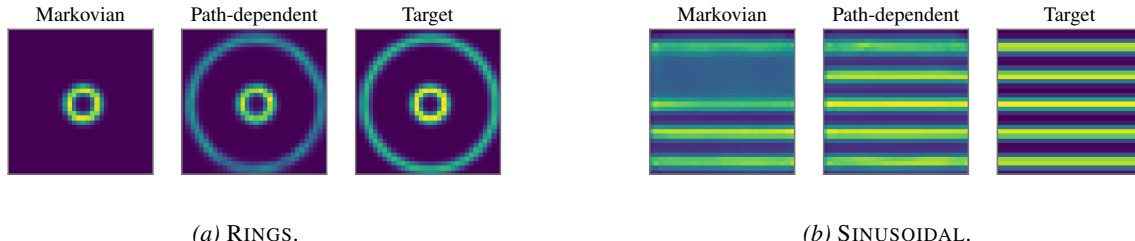

*(a)* RINGS.

*(b)* SINUSOIDAL.

*Figure 12.* **Results for Lazy Random Walk**. A path-dependent parameterization yields a better distributional approximation than a Markovian one for both the RINGS (left) and SINUSOIDAL (right) targets, which we describe in Equations (21) and (22).

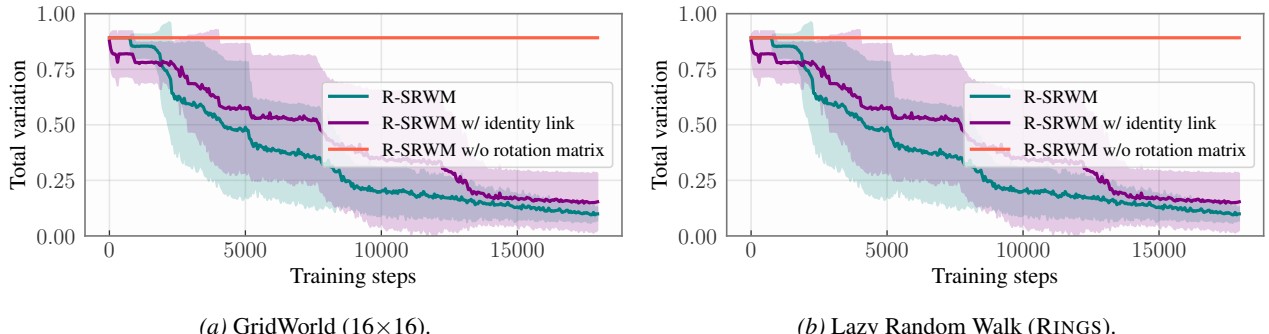

*(a)* GridWorld (16×16).

*(b)* Lazy Random Walk (RINGS).

*Figure 13.* **The rotation matrix plays a key role in accelerating learning convergence**. While both the chosen link function $\sigma$ (Equation (23)) and the rotation matrix $R$ are beneficial for training, removing $R$ effectively hinders training.

**Lazy Random Walk.** We evaluate our algorithm on the Lazy Random Ralk (LRR) task, which has also been considered by Dall'Antonia et al. (2026). Similarly to the GridWorld, the sampler starts at $\mathbf{0} \in \mathbb{R}^d$ and, at each step, updates its current state $\mathbf{s}$ as $\mathbf{s} \leftarrow \mathbf{s} + \mathbf{v}$, with $\mathbf{v} \in \{\mathbf{e}_1, \ldots, \mathbf{e}_d, \mathbf{0}\}$ and $\mathbf{e}_i$ as the $i$-column of the $d$-dimensional identity matrix. In contrast to the GridWorld, however, the generative process does not stop when the transition corresponding to $\mathbf{0}$ is realized; instead, it proceeds for exactly $T$ steps. From a fundamental standpoint, this ensures the underlying stochastic process is aperiodic, and we empirically found that such a generative process is less prone to collapse than the one in Figure 11a. In this context, we augmented the state $\mathbf{s}$ with the current time-step $t$, $(\mathbf{s}, t)$, and restricted the sampler to $[-M, M]^d$. As such, the terminal set is $\mathcal{X} = \{(\mathbf{s}, T) \colon \mathbf{s} \in [-M, M]^{15}\}$. We parameterized $t$ with Fourier's positional encoding (Vaswani et al., 2023). Additionally, we considered the RINGS and SINUSOIDAL target distributions, depicted in Figure 12(a-b) (right), defined by

$$R_{\text{Rings}}(\mathbf{s}) = 0.6 \cdot \exp\{-10(\|\tilde{\mathbf{s}}\| - 0.2)^2\} + 0.3 \cdot \exp\{-10(\|\tilde{\mathbf{s}}\| - 0.9)^2\}, \tag{21}$$

in which $\tilde{\mathbf{s}} = M^{-1}\mathbf{s} \in [-1, 1]$ and $\| \cdot \|$ is the Euclidean norm, and

$$R_{\text{Sinusoidal}}(\mathbf{s}) = \cos(\mathbf{s}_2). \tag{22}$$

Intuitively, both target distributions require the sampler to perform significantly different actions at similar states. Under this condition, as we illustrated in Figure 5, learning convergence is hampered by state aliasing. Results for the LRR task are shown in Figure 12, confirming the higher-quality goodness-of-fit of our path-dependent parameterization.

**Ising model.** We further evaluate our approach in the simulation of the Ising model, which is a common benchmark for discrete sampling in the literature (e.g., (Zhang et al., 2022; Liu et al., 2024)). Simply put, the state space is $\{-1, 1\}^d$, with $-1$ and $1$ representing a particle's spin, and the (unnormalized) target distribution is

*Table 5.* FCS for the Ising model simulation.

| $d$ | Markovian | non-Markovian |
|---|---|---|
| 36 | $0.15_{\pm 0.03}$ | $\mathbf{0.06}_{\pm 0.01}$ |
| 64 | $0.20_{\pm 0.02}$ | $\mathbf{0.08}_{\pm 0.01}$ |

$$\log R(\mathbf{x}) = \mathbf{x}^T \mathbf{h} + \frac{1}{2}\mathbf{x}^T \mathbf{J} \mathbf{x},$$

with $\mathbf{x} \in \{-1, 1\}^d$ and $\mathbf{h} \in \mathbb{R}^d$ and $\mathbf{J} \in \mathbb{R}^{d \times d}$ fixed. Intuitively, $-1/2\mathbf{x}^T\mathbf{J}\mathbf{x}$ represents the energy associated to the particles' pairwise interactions, and $-\mathbf{x}^T\mathbf{h}$ measures the energy from an external magnetic field. We let $\mathbf{J} = \mathbf{v}\mathbf{v}^T - \text{diag}(\mathbf{v}\mathbf{v}^T)$ for

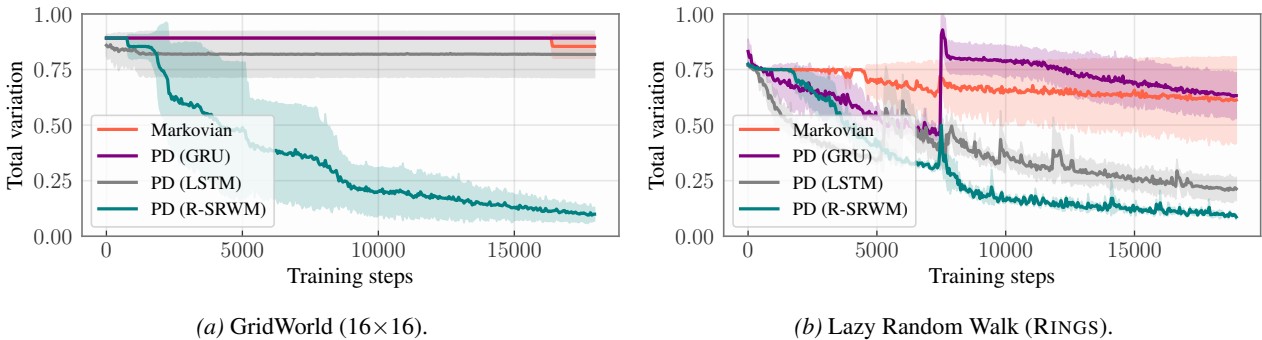

*(a)* GridWorld (16×16).   *(b)* Lazy Random Walk (RINGS).

*Figure 14.* **Our SRWM leads to faster convergence than gated RNNs.** In contrast to LSTM and GRU, our SRWM-based policy network promotes the diversity of the model's hidden representations—via the rotation matrix—reducing aliasing and accelerating convergence.

$\mathbf{v} \sim \mathcal{U}(\{\mathbf{v} \in \mathbb{R}^d \colon \|\mathbf{v}\|_2 = 1\})$, representing a ferromagnetic system (i.e., the pairwise-interaction energy is minimized when the particles are spin-aligned), and $\mathbf{h} \sim \mathcal{N}(\mathbf{0}, \mathbf{I})$. In Table 5, we selected $d \in \{36, 64\}$ and averaged the FCS for both the Markovian and path-dependent samplers across three independent runs. Again, we observed our non-Markovian parameterization to find a better approximation to the target distribution than its Markovian counterpart.

### F.3. Ablation studies

**Rotation matrix & link function.** Our parameterization for the learned latent dynamical system, based on a self-referential weight matrix, has two primary components: a rotation matrix ($R$) and a link function ($\sigma$). As in (Irie et al., 2022), we set

$$\sigma(\mathbf{x}) = \frac{\mathrm{ELU}(\mathbf{x}) + 1}{\mathbf{1}^T \mathrm{ELU}(\mathbf{x}) + d} \tag{23}$$

for $\mathbf{x} \in \mathbb{R}^d$ and $\mathbf{1}$ as the $d$-dimensional vector of ones. (ELU, applied element-wise, is defined by $x \mapsto x$ if $x \geq 0$ and $x \mapsto \exp\{x\} - 1$ if $x < 0$). To better understand the effect of these design choices over the effectiveness of our path-dependent parameterization, we show in Figure 13 the TV distance of our model after (i) removing the rotation matrix and (ii) setting an identity link function. We consider the Lazy Random Walk (RINGS) and GridWorld (Figure 7) domains, for which this metric can be tractably computed. Remarkably, both experiments confirm that the rotation matrix plays a pivotal role in accelerating learning convergence—and that the link function, despite beneficial, is of relatively little importance. This corroborates our mechanical intuition that, by rotating the underlying weight matrix's eigenvectors, the rotation matrix effectively mitigates state aliasing.

**Comparison against gated RNNs.** As mentioned, we compared our modified SRWM against conventional gataed RNNs, including LSTM (Hochreiter & Schmidhuber, 1997) and GRU (Cho et al., 2014), for parameterizing the latent dynamical system, and found our SRWM to be far more effective in speeding up learning convergence in many tasks. We illustrate this in Figure 14. The reason for this is that, differently from LSTM and GRU, our SRWM has a built-in mechanism—the rotation matrix, as explained—for mitigating state aliasing.

