# OpenReview forum: "Path-dependent Discrete Amortized Inference"
_ICML.cc/2026/Conference — ICML 2026 spotlight_

### Official Review · Reviewer_H1vg · 2026-02-26

**Soundness:** 3
**Presentation:** 3
**Significance:** 3
**Originality:** 3
**Overall Recommendation:** 5
**Confidence:** 3

**Summary:**

The goal is to sample from a distribution on a discrete space $\pi(x) = \frac{R(x)}{Z}$ when only its unnormalized pdf $R(x)$ is known.

The authors base themselves on the framework of GLowNets. The state space is viewed as nodes connected by edges (pointed DAG) and the sampling process consists in running a Markov Chain with a transition probability $p(x_{t+1} | x_t)$. This transition probability is parameterized by a neural network and learnt by minimizing a cost function (Eq 2) that enforces that its stationary distribution is $\pi(x)$.

The authors' contribution consists in extending this formalism to an *augmented* state space $(x, a)$ where $a$ is a latent variable (see Figure 1 and Section 4). It can be intuitively interpreted as storing the *history* of visited states, like the memory variable of a Recurrent Neural Network or the momentum variable of the Hamiltonian Monte Carlo algorithm. The authors similarly equip this augmented state space $(x, a)$ with a graph structure, and run a Markov Chain on it with a learnable transition probability, and adapt the cost function so that the stationary distribution of $x$ is $\pi(x)$.

The authors claim that augmenting the state space with this history variable $a$ enables more flexible parameterizations of the Markov Chain and accelerates its convergence. They verify this claim on simple examples (section 3.1), simple experiments (Figures 3.1 and 5), as well as other experiments (see section 5).

**Compliance With Llm Reviewing Policy:**

Affirmed.

**Final Justification:**

This is a neat contribution and I am happy to maintain my score and recommend acceptance.

**Key Questions For Authors:**

Q1. Could the authors include an experiment where we can explicitly see that augmenting the state space helps discover more modes? For example, by reporting the number of discovered modes by GFlowNet vs. Augmented GFlowNet as a function of the iterations.

**Limitations:**

Yes

**Strengths And Weaknesses:**

**Strengths**: the paper is clearly written, and the authors verify the usefulness of augmenting the state space of the Markov Chain on pedagogical examples (Lines environment and linear policies) --- this is appreciated.

**Weaknesses**: while I am familiar with the sampling literature, I was not so familiar with the GFlowNet literature and found some of the parts very detailed and hard to follow, in particular Proposition 3.1, sections 3.2 and 4.2.

---

> ### Author Rebuttal · Authors · 2026-03-30
>
> We are thankful for the reviewer’s feedback and support for our work. We will happily include a background on GFlowNets in the supplement, connecting it to the broader sampling literature (e.g., VI [1] and diffusion samplers [2]).
>
> > Q1. Could the authors include an experiment where we can explicitly see that augmenting the state space helps discover more modes?
>
> Sure. We have included a plot of the average unnormalized target ($R(x)$) of the most valuable observed samples vs. the number of training iterations for the hypergrid and sequence generation tasks.
> This metric is conventionally used for measuring mode discovery speed of amortized samplers. Results can be consulted [here](https://anonymous.4open.science/r/pdgflownets-1334/mode_discovery.png).
> Notably, our approach leads to faster mode discovery than a traditional Markovian model for these tasks.
>
> Intuitively, the reason for this is that our path-dependent parameterization boosts the expressivity of the policy network (as shown in Section 3), which is less likely to get stuck at certain high-probability regions of a multi-modal target distribution.
>
>
> We will include this experiment into the revised manuscript. Thanks for the feedback!
>
>
> [1] GFlowNets and variational inference, Malkin et al., ICLR 2023.
>
> [2] Unifying Generative Models with GFlowNets and Beyond, Zhang et al., ICML 2022 Beyond Bayes workshop.

---

> > ### Author Rebuttal · Reviewer_H1vg · 2026-04-02
> >
> > Thanks. I appreciate the authors including the mode-discovery results in the revised manuscript. This is a neat contribution and I am happy to maintain my score and recommend acceptance.

---

### Official Review · Reviewer_9dYi · 2026-03-05

**Soundness:** 3
**Presentation:** 3
**Significance:** 3
**Originality:** 3
**Overall Recommendation:** 5
**Confidence:** 4

**Summary:**

Learning Markov decision processes (MDPs) offers a solution to sampling from intractable distributions over compositional or otherwise discrete variables. In a similar spirit to how Langevin and Hamiltonian dynamics have been used to yield better MCMC algorithms---by introducing an auxiliary variable---the authors introduce a latent variable to their MDP whose dynamics are determined by a learnable update rule parameterised by neural networks. The introduction of such a latent variable destroys the Markov (memorylessness) property of the decision process, and allows the distribution at a particular timestep to depend on the full sampling trajectory. The authors demonstrate both theoretically and empirically that this leads to strictly more expressive sampling procedures.

**Compliance With Llm Reviewing Policy:**

Affirmed.

**Final Justification:**

The authors' rebuttal reinforced my prior assessment of the paper. It would be a valuable contribution to the field and I hope to see it accepted.

**Key Questions For Authors:**

Q1.) Is SRWM an instance of a recurrent policy network?

Q2.) Could the authors highlight to me the way in which their approach is *amortised*?

**Limitations:**

The authors could perhaps discuss the limitations of their work further.

**Strengths And Weaknesses:**

### Strengths

- The narrative, structure, and writing quality are all solid. I like that figures 1-5 are included early in the exposition to help to convey the narrative rather than reserving them for the experiments section, particularly figure 3.
- The idea seems sensible and with clear theoretical and intuitive advantages over Markovian samplers.
- The improvement over Markovian samplers is demonstrated comprehensively by the empirical results.
- The theory seems sound to me (though I am not an expert in this area).
- The appendices provide numerous extensions and further discussion for particularly interested readers. I think this is great, and often undervalued by reviewers.

### Weaknesses

1. All but two (SIX6 and PHO4) of the experimental settings are synthetic. I don't think this is a major weakness as both the synthetic experiments and theoretical results are highly comprehensive, but a greater variety of real-world experiments would nonetheless bolster the paper.

2. The experimental details are somewhat on the thin side. It would be great if the authors could supplement these details into the appendix in their next revision. It's great that the code is provided, but readers should not have to parse an entire codebase in order to extract some minor experimental detail that they might be wondering about.

### Nit-Picks:
(These are not criticisms, the purpose is to help the authors in tidying up the manuscript for revision)

a.) In the 3rd line of the 2nd paragraph of the introduction: "low dimension" should be changed to "low dimensional settings" or "low dimensions".

b.) The axis labels of Figure 4b) are occluded/missing and this makes it difficult to understand the graph.

c.) The phrase "a latent dynamics" doesn't really make sense, but the authors use it many times. "dynamics" is a plural word, so we can say that "the dynamics *are*...". Otherwise, I recommend talking about the "latent dynamics mechanism" or "latent variable update rule" to refer to the neural net that governs how $W_t$ evolves.

d.) "FCS" is defined in appendix C, but is used as the performance metric in some experiments. I think it's fine to defer the mathematical definition to an appendix, but what the acronym stands for certainly needs to be introduced in the main part of the paper.

e.) In the "Sequence design" paragraph of section 5, "results are consistency" should be changed to "results are consistent".

### My Take

Overall, I think the paper is good and that the ICML community would benefit from this paper's appearance. However, I am not an expert in the paper's domain so may have missed some important weaknesses.

---

> ### Author Rebuttal · Authors · 2026-03-30
>
> We are thankful for the reviewer’s supportive and detailed feedback. We will incorporate the additional discussions and suggested changes into the revised manuscript.
>
> > All but two (SIX6 and PHO4) of the experimental settings are synthetic.
>
> We acknowledge this. The primary objective of our work was to pin down the issue of state aliasing in amortized sampling through stylized constructions (Section 3) and introduce a general-purpose solution (Section 4) for this via state space lifting.
>
> In particular, our experimental analysis was built on standard benchmark problems in the amortized sampling literature. However, to provide further empirical evidence, we have also included results for the simulation of the 2D Ising model on a $6 \times 6$ ($36$ particles) and $8 \times 8$ ($64$ particles) grid, which is commonly used for benchmarking (e.g., [1]).
>
> | Model | Grid Size | FCS |
> |-------|-----------|-------|
> | Markovian | 36 | 0.15 ± 0.03 |
> | Markovian | 64 | 0.20 ± 0.02 |
> | Path-dependent | 36 | 0.06 ± 0.01 |
> | Path-dependent | 64 | 0.08 ± 0.01 |
>
>
> Results were averaged across three independent runs. We will include this experiment in the revised manuscript.
>
> [1] Generative Flow Networks for Discrete Probabilistic Modeling, Zhang et al. ICML 2022.
>
>
> > The experimental details are somewhat on the thin side.
>
> Thanks for the suggestion! We will revise the experimental details section to add further details on our empirical setup in the updated text.
>
> > Nit-Picks
>
> We appreciate your suggestions. We will fix all typos, update “latent dynamics” to “latent dynamics mechanism”, and bring the definition of FCS to the main text.
>
> > Is SRWM an instance of a recurrent policy network?
>
> Yes, as SRWMs are a specific class of recurrent neural networks, a policy function parameterized by a SRWM can be seen as an instance of a recurrent policy network.
>
> > Could the authors highlight to me the way in which their approach is amortised?
>
> Thanks for the question. Our approach is amortized in the sense that the forward policy is learned as a parametric function from the state space $\mathcal{S}$ to a probability distribution over $\mathcal{S} \cup \mathcal{X}$, with $\mathcal{X}$ being the set of terminal states. That is, we amortize the policy over $\mathcal{S}$.
>
> This is slightly different from the conventional terminology in the VI/VAE literatures, for which amortization occurs at the data-level. However, the term (as described above) has found wide adoption in the sampling literature (e.g., [1, 2]), which is the main reason we use it in our work.
>
> [1] Amortizing intractable inference in large language models, Hu et al., ICLR 2024.
>
> [2] Action abstractions for amortized sampling, Boussif et al., ICLR 2024.
>
> ---
>
> We are glad for your careful review of our work! As mentioned, we will incorporate all your suggestions into the revised manuscript.

---

> > ### Author Rebuttal · Reviewer_9dYi · 2026-04-01
> >
> > I thank the authors for their comprehensive response to my review. I'm pleased to hear about the promised changes, and I'm very grateful for the authors' patience in answering my "Key Questions". I'm particularly impressed to see the inclusion of a further benchmarking setting.
> >
> > With all my concerns fully addressed, I maintain my favourable score but with renewed confidence (3->4). I hope to see the paper accepted, and I congratulate the authors on their nice work!

---

### Official Review · Reviewer_bWvb · 2026-03-12

**Soundness:** 3
**Presentation:** 3
**Significance:** 3
**Originality:** 3
**Overall Recommendation:** 5
**Confidence:** 4

**Summary:**

The paper tackles sampling in compositional spaces by “lifting” the usual Markovian MDP with a learnable latent dynamics in GFlowNets so the policy can depend on the full trajectory. The authors argue that Markovian assumptions impede credit assignment and expressivity (via state aliasing). They formalize a path-dependent framework, adapt TB/SubTB/CB learning objectives to the lifted space, and propose a recurrent self referential weight matrix policy. Theory shows strictly improved expressivity over Markovian parameterizations (linear, GNN, ReLU cases). Empirically, across LINES, grid world, set/sequence generation, and other benchmarks, the approach often converges faster and matches targets more closely than Markovian baselines.

**Compliance With Llm Reviewing Policy:**

Affirmed.

**Final Justification:**

The authors' rebuttal addressed my main concerns. I keep my acceptance decision as before.

**Key Questions For Authors:**

How do SRWM policies compare against GRU/LSTM/Temporal Transformer policies when controlling for parameters and compute across all benchmarks? Can you provide a full ablation on the rotation R, the gating \sigma, and latent dimension d?

What is the training/inference overhead (FLOPs, memory, wall clock) versus Markovian GFlowNets for comparable performance? How does cost scale with trajectory length and latent size?

How sensitive are results to the choice of exploratory policy and its schedule (e.g., ϵ-greedy vs. temperature controlled)? Any signs of mode dropping or collapse under different?

Section D sketches a stochastic extension. Do preliminary experiments indicate accuracy or stability benefits, and what are the practical challenges (e.g., variance, continuous state handling)?

In Proposition 4.8, what happens empirically if p_B is also path dependent? Do we still recover near Markovian p_F on the unlifted DAG, or does it materially change learning dynamics?

**Limitations:**

•	The analysis and guarantees hinge on deterministic updates to keep the lifted space countable; the stochastic variant is relegated to an appendix and left for future work.
•	Certain results (e.g., Proposition 4.8) assume a Markovian backward policy on the unlifted graph; practical impact of violating this assumption is not fully explored.
•	The approach is framed for DAG like generation; extending to cyclic/non acyclic settings is discussed but not demonstrated empirically.

**Strengths And Weaknesses:**

Strong points
•	The paper pinpoints how Markovian policies in GFlowNets suffer from state aliasing and weak signal propagation, motivating path dependence.
•	Propositions show strict expressivity gains over Markovian models for linear policies (Proposition 3.1) and for GNN based policies (Proposition 3.2), with proofs and illustrative constructions
•	TB, SubTB, and CB are extended with guarantees that the marginal over terminal states matches the target, preserving the core GFlowNet correctness.
•	The modified SRWM provides a principled, efficient way to maintain trajectory dependent memory, with a rationale for why it mitigates the socalled state aliasing.
•	Results on LINES, grid world, set/sequence generation (incl. SIX6/PHO4), preference learning, and lazy random walk consistently favor the path dependent parameterization in TV/FCS and convergence.

Weak points
•	While Markovian baselines are included, stronger non Markovian alternatives (e.g., LSTM/GRU/Transformer policies with sophisticated credit-assignment or exploration strategies) are only briefly mentioned and not systematically benchmarked.
•	The SRWM choice is motivated, but more granular ablations (e.g., varying latent size or comparing to gated RNNs under matched parameter counts) would clarify where gains arise
•	Compute/complexity accounting. The lifted state introduces extra computation and memory; a rigorous wall clock/throughput comparison and per sample cost analysis is missing.

---

> ### Author Rebuttal · Authors · 2026-03-30
>
> We appreciate the reviewer's thoughtful feedback and support for our work. Below, we address each of your concerns. We will incorporate all additional experiments and discussions into the revised manuscript.
>
> > Ablations and comparisons against gated RNNs.
>
> Thanks for the questions. We (1) benchmarked our model (henceforth, R-SRWM) against gated RNNs, and (2) measured its performance after (i) removing the rotation matrix and (ii) using an identity link function.
>
> Results for (1) can be found [here](https://anonymous.4open.science/r/pdgflownets-1334/hypergrids_gated_rnns.png) and [here](https://anonymous.4open.science/r/pdgflownets-1334/lazy_random_walk_gated_rnns.png) for the hypergrid and lazy random tasks, respectively.
> Similarly, results for (2) can be found [here](https://anonymous.4open.science/r/pdgflownets-1334/hypergrids_ablations.png) and [here](https://anonymous.4open.science/r/pdgflownets-1334/lazy_random_walk_ablations.png).
> R-SRWM outperforms both traditional gated RNNs and a conventional (unrotated) SRWM for these tasks.
> Although both the link function ($\sigma$) and the rotation matrix ($R$) play a key role in this performance, we found that $R$ is especially beneficial, which is consistent with our intuition regarding state aliasing.
>
> We also ran additional experiments with varying size for the latent [dimension](https://anonymous.4open.science/r/pdgflownets-1334/dimension_ablations_hypergrids.png) $d$, $\epsilon$ for $\epsilon$-[greedy policies](https://anonymous.4open.science/r/pdgflownets-1334/ablation_epsilon_greedy.png), and $T$ for [temperature-controlled exploration](https://anonymous.4open.science/r/pdgflownets-1334/ablation_tempered_policies.png), but found no discernible pattern.
> Despite being more expressive, larger models are harder to train and have a larger per-step computational cost.
> The design of scaling laws for amortized sampling, nevertheless, remains an interesting venue for future research.
> Similarly, small values of $\epsilon$ (resp. $T$) lead to poor exploration and slow training convergence, while large values may hamper the sampler's ability to explore high-probability regions. We will happily include these experiments into the revised manuscript.
>
>
> > Computational complexity measurement.
>
> Thank you for the questions. Simulating the latent dynamical system adds a small computational overhead to training and inference, albeit it may reduce overall training time. We present the number of FLOPs for both the Markovian and non-Markovian samplers below.
>
> | Task | Method | Training FLOPs/step | Inference FLOPs/batch |
> |------|--------|--------------------|-----------------------|
> | Hypergrid | Markovian | 6.46e6 | 2.16e6 |
> | Hypergrid | Path-dependent | 7.62e6 | 2.97e6 |
> | Lazy random walk | Markovian | 1.29e7 | 4.27e6 |
> | Lazy random walk | Path-dependent | 1.49e7 | 5.08e6 |
>
> Regarding complexity, simulation cost is dominated by the $\mathcal{O}(d^{3})$ complexity of matrix multiplication with dimension $d$. To mitigate this, we recommend using a small latent dimension (e.g., 32), later expanded in the policy’s network last layer. We will clarify this in the text. Also, as trajectories are iteratively generated, complexity for both the Markovian and our path-dependent samplers is $\mathcal{O}(L)$, with $L$ being the trajectory length.
>
> > Stochastic extension.
>
> We are glad for the opportunity to further elaborate on the stochastic extension of our method. We have not yet carried out experiments with a stochastic latent dynamical system, as our discussion in Section D was primarily focused on the theoretical viability of implementing such a sampler.
>
> The practical challenges in doing so, in particular, are two-fold: (i) designing a terminal distribution over latents and (ii) defining a computationally amenable and sufficiently expressive policy network. We are currently investigating these directions.
>
> > In Proposition 4.8, what happens empirically if p_B is also path dependent?
>
> We appreciate your input. When $p_{B}$ is path-dependent, $p_{F}$’s near-Markovianity is not guaranteed. To understand this, we considered the set generation task. We measured, for a random selection of incomplete states $\{s_{1}, \dots, s_{N}\}$, the maximum KL divergence between forward policies over past trajectories, i.e.,
> $$
> 	\mathrm{KL}(s) = \max_{\tau \colon s_{o} \rightsquigarrow s, \tau’ \colon s_{o} \rightsquigarrow s} \mathrm{KL}[p_{F}(\cdot | \tau) || p_{F}(\cdot | \tau’)].
> $$
> If $p_{F}$ is Markovian, $\mathrm{KL}(s) = 0$.
> We report $\frac{1}{N} \sum_{1 \le i \le N} \mathrm{KL}(s_{i})$. As we can observe [here](https://anonymous.4open.science/r/pdgflownets-1334/markovian_vs_non_markovian_pb.png), $\frac{1}{N} \sum_{1 \le i \le N} \mathrm{KL}(s_{i})$ only asymptotically vanishes when $p_{B}$ is Markovian (uniform). This corroborates Proposition 4.8.
>
> ---
>
> We are thankful for your valuable input! We will include all novel experiments into the updated text.

---

> > ### Author Rebuttal · Reviewer_bWvb · 2026-04-04
> >
> > Thanks for providing more details and results. I will keep my current score.

---

### Decision · Program_Chairs · 2026-04-30

**Decision:**

Accept (spotlight)

**Comment:**

This paper studies discrete amortized inference in compositional spaces where the standard Markovian assumption in existing amortized samplers can limit both expressivity and training signal propagation. To address this, the authors propose a path-dependent formulation that augments the state with learnable latent dynamics, allowing the policy to depend on the full trajectory rather than only the current state. Reviewers agree that this is a technically interesting and well-motivated contribution, with clear connections to both the GFlowNet literature and broader sampling methodology. Reviewers find the central idea sensible, the theoretical framing sound, and the empirical results convincing. Given the consistently positive reviews, the solid theoretical and empirical support, and the effective rebuttal, acceptance is recommended. The final version would benefit from incorporating the promised clarifications, ablations, and added experimental details.